# Probabilistic Analysis of Future Drought Propagation, Persistence, and Spatial Concurrence in Monsoon-Dominant Asian Region under Climate Change

Dineshkumar Muthuvel[1], Xiaosheng Qin[1]

[1] School of Civil and Environmental Engineering, Nanyang Technological University,
50 Nanyang Ave, Singapore 639798

*Correspondence to*: Xiaosheng Qin (xsqin@ntu.edu.sg)

**Abstract.** This study examines future drought propagation (the temporal transition from meteorological to agricultural droughts), persistence (inter-seasonal agricultural droughts), and spatial concurrence (simultaneous occurrence of monsoonal agricultural droughts across regions) under climate change using a multivariate copula approach in Monsoon-dominant Asia. Standardized Precipitation Index (SPI) and Standardized Soil moisture Index (SSI) are used to analyse meteorological and agricultural droughts, respectively. Under the worst-case emission scenario (SSP5-8.5), South Asia (excluding Western and Peninsula India) and Eastern China are projected to experience intensified drought propagation compared to the historical period (1975-2014). In addition to increased propagation in these regions, the propagated agricultural droughts are expected to persist across seasons in the future. At the hydrologically significant Tibetan Plateau, all-season droughts that were historically rare, with return periods exceeding 50 years, could occur as frequently as once every 5 years in the far-future period (2061-2100). Random Forest models indicate that the temperature is a key driver of future agricultural droughts in nearly half of the study area. The increasing non-rainfall-related agricultural droughts in the far-future could be attributed to the rise in temperature. Based on bivariate return periods of spatial concurrence, frequent future spatial drought concurrence is anticipated between populous South Asia and East Asia compared to the historical timeframe, posing risks to water and food security. Conversely, Southeast Asia is projected to experience reduced spatial drought concurrence with other regions, which could encourage greater regional cooperation. Overall, this comprehensive approach which integrates three aspects of drought dynamics, offers valuable insights for climate change mitigation, planning, and adaptation.

**(a) Drought propagation using conditional probability**

**(c) Spatial drought concurrence using joint and conditional probabilities**

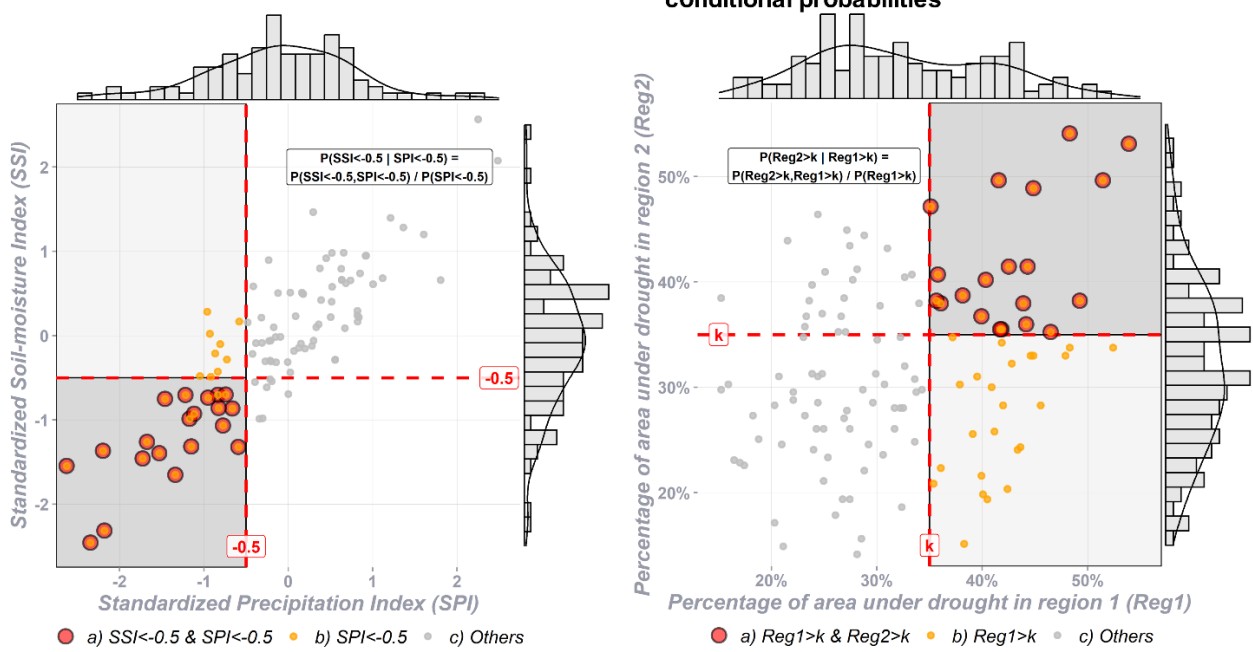

$$P(SSI<-0.5 \mid SPI<-0.5) = P(SSI<-0.5, SPI<-0.5) / P(SPI<-0.5)$$

- a) SSI<-0.5 & SPI<-0.5
- b) SPI<-0.5
- c) Others

$$P(Reg2>k \mid Reg1>k) = P(Reg2>k, Reg1>k) / P(Reg1>k)$$

- a) Reg1>k & Reg2>k
- b) Reg1>k
- c) Others

**(b) Drought persistence across three seasons (pre-monsoon, monsoon, and post-monsoon seasons)**

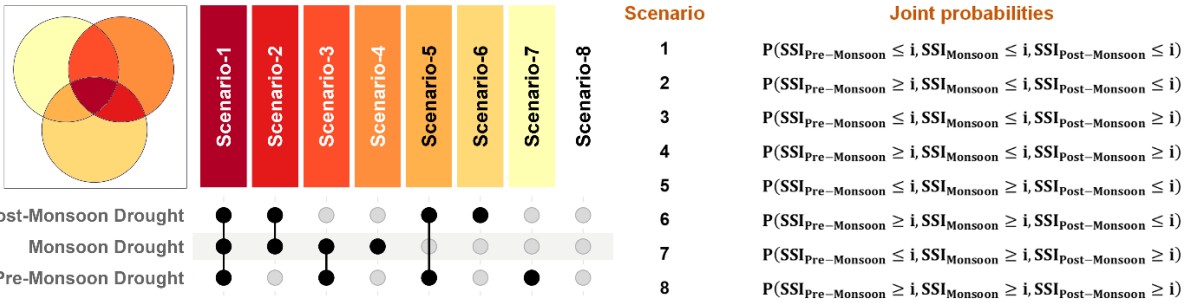

| Scenario | Joint probabilities |
|---|---|
| 1 | $P(SSI_{Pre-Monsoon} \leq i, SSI_{Monsoon} \leq i, SSI_{Post-Monsoon} \leq i)$ |
| 2 | $P(SSI_{Pre-Monsoon} \geq i, SSI_{Monsoon} \leq i, SSI_{Post-Monsoon} \leq i)$ |
| 3 | $P(SSI_{Pre-Monsoon} \leq i, SSI_{Monsoon} \leq i, SSI_{Post-Monsoon} \geq i)$ |
| 4 | $P(SSI_{Pre-Monsoon} \geq i, SSI_{Monsoon} \leq i, SSI_{Post-Monsoon} \geq i)$ |
| 5 | $P(SSI_{Pre-Monsoon} \leq i, SSI_{Monsoon} \geq i, SSI_{Post-Monsoon} \leq i)$ |
| 6 | $P(SSI_{Pre-Monsoon} \geq i, SSI_{Monsoon} \geq i, SSI_{Post-Monsoon} \leq i)$ |
| 7 | $P(SSI_{Pre-Monsoon} \leq i, SSI_{Monsoon} \geq i, SSI_{Post-Monsoon} \geq i)$ |
| 8 | $P(SSI_{Pre-Monsoon} \geq i, SSI_{Monsoon} \geq i, SSI_{Post-Monsoon} \geq i)$ |

# 1 Introduction

Droughts, though not instantaneous, rank among the costliest disasters due to their prolonged and widespread impacts (Hao et al., 2016; Smith and Katz, 2013; Smith and Matthews, 2015). They are known to manifest in different forms, beginning with meteorological drought caused by a precipitation deficit. If sustained, this can reduce runoff and soil moisture, leading to hydrological and agricultural droughts that impact entire ecosystems (Das et al., 2022). The propagation process is driven by multiple factors such as regional climate, teleconnections, topography, and anthropogenic activities (Han et al., 2019). Despite its complexity, understanding the progression of meteorological droughts into other forms is crucial for enhancing disaster preparedness and mitigation, particularly in the context of climate change. When a meteorological drought evolves into an

agricultural or hydrological drought, the risks are further heightened if the agricultural drought persists across multiple seasons (Ford and Labosier, 2014). Given the spatial-temporal nature of droughts, such events can also occur simultaneously across multiple regions (Gaupp et al., 2017). Therefore, a comprehensive assessment requires examining the characteristics of drought propagation, persistence, and spatial concurrence.

The different drought forms are related to the components and processes of the hydrological cycle. The propagation of different forms includes a certain time lag, which most studies focus on (Barker et al., 2016; Bevacqua et al., 2021; Wang et al., 2024). Such studies use the cross-correlation between time series of monthly soil moisture (or streamflow) and accumulated precipitation to deduce propagation time. Climate characteristics are known to influence drought propagation. For instance, Zhang et al. (2021) found that arid basins tend to have shorter propagation durations than humid and sub-humid basins. Seasonality also influences propagation, and Dai et al. (2022) determined faster propagation in summer and autumn compared to spring and winter. Apart from climatic factors, region specific characteristics of soil texture, vegetation, and topography affect evapotranspiration and eventually the propagation duration (Wu and Hu, 2024). Hence, propagation duration estimated using the correlation-based approach captures the interplay between the key factors. Further, propagation probability measuring the likelihood of meteorological drought evolving to other forms are computed using copula functions (Xu et al., 2021). Analysing the propagation process helps in forecasting impending agricultural droughts, which is pertinent since soil moisture deficits (i.e., agricultural droughts) directly impact crop and vegetation growth (Modanesi et al., 2020). After establishing the relationship between precursor meteorological and antecedent agricultural droughts, the persistence and cross-regional concurrence of agricultural droughts need to be analysed.

Ford and Labosier (2014) defined persistence as the tendency of droughts to extend temporally from one season to another. Inter-seasonal drought dynamics can also be analysed using a copula-based multivariate approach, treating drought index values from each season as random variables (Chen et al., 2016; Fang et al., 2019; Shi et al., 2020; Swain et al., 2024; Xiao et al., 2017). These previous studies analysed transition from dryness to wetness (and vice versa), prolonged wetness and prolonged dryness between two successive seasons using bivariate copulas. Of the four possible inter-seasonal scenarios, prolonged dryness referring to inter-seasonal drought persistence can affect vegetation and crop growth. Nonetheless, limiting the analysis to two consecutive seasons may overlook long-term drought persistence across seasons that overlap with crop growth cycles. Crops are cultivated across multiple seasons throughout the study area, with different stages of crop growth aligning with the pre-monsoon, monsoon, and post-monsoon seasons. For example, in China, spring crops are typically sown in May and harvested around October, while winter crops like wheat are planted in September and harvested by following June (Li and Lei, 2021). Similarly, in South Asia, different crops are grown in three different phases including Zaid, Kharif, and Rabi, that correspond to the pre-monsoon, monsoon, and post-monsoon seasons, respectively (Joseph and Ghosh, 2023). Adequate soil moisture during all these periods is critical for crop development, and agricultural droughts affecting the three phases could be highly detrimental. Beyond temporal aspects, the spatial concurrence (or compounding) of droughts poses a significant threat to food or energy security when events occur simultaneously across multiple breadbaskets or hydropower

basins (Lv et al., 2025, 2024). In this context, studies have utilized copulas to estimate the multivariate joint probability of key crop and pasture regions experiencing concurrent droughts (Gaupp et al., 2020; Sarhadi et al., 2018).

Despite extensive research on drought-related topics, several gaps remain to be addressed. First, various aspects of droughts, such as propagation, persistence, and spatial concurrence, have largely been studied in isolation. Several seminal works (Dai et al., 2022; Ding et al., 2021; Fawen et al., 2023; Xu et al., 2023) focus on the propagation process, but tend to overlook its repercussions, namely the persistence and spatial concurrence of agricultural droughts. While propagation analysis deals with the transition from a precursor meteorological to an agricultural drought, persistence and spatial concurrence capture their temporal and spatial extents. Given the spatiotemporal nature of droughts, it is crucial to examine these three aspects together for a more comprehensive assessment. Secondly, previous studies have considered drought persistence as a phenomenon spanning two consecutive seasons. However, given the importance of sufficient soil moisture across three sequential seasons (pre-monsoon, monsoon, and post-monsoon) for crop growth, a trivariate extension of the existing bivariate copula framework could more accurately capture drought persistence. Finally, most studies on drought persistence and spatial concurrence reply on observational data. Incorporating climate projection data can offer insights into how droughts may persist and concur spatially in the future under climate change.

To address the aforementioned gaps, this study proposes a comprehensive copula-based multivariate probabilistic framework, which will be applied to climate model projection data. The approach is applied to the monsoon-dominant Asian region under climate change, comparing drought characteristics between future and historical timeframes. The monsoon-dominant Asian region is home to several global rice bowls and wheat baskets that support local food security and play a key role in the international food trade (Gaupp et al., 2020). Given the region's significance, it is appropriate to study a comprehensive drought framework within this study area. Initially, the propagation probability from meteorological to agricultural droughts is estimated for different seasons, and the influencing factors are identified. Next, the persistence of agricultural droughts across seasons is evaluated. Finally, the spatial concurrence of monsoonal agricultural droughts across various regions is assessed. This study offers a novel attempt to integrate these three aspects of droughts and analyse their interrelationships using multivariate copula functions. The resulting framework will identify key regions and patterns of future droughts, providing valuable insights for mitigation strategies.

## 2 Methodology

### 2.1 Drought propagation

#### 2.1.1 Drought propagation duration

Indices such as Standardized Precipitation Index (SPI) and the Standardized Soil Moisture Index (SSI) are standard normal variates commonly used to quantify meteorological and agricultural droughts, respectively (Hao and AghaKouchak, 2013). SPI is based on monthly precipitation, while SSI relies on monthly soil moisture. These standardized indices are widely

applied in drought studies due to their flexibility in capturing seasonal variations through different timescales (TS). Timescales represent accumulation periods, defined by aggregating precipitation or soil moisture over a specified number of months. For each month, the accumulated values (for a given timescale) are transformed into a standardized index. The standardization process involves the computing the cumulative probability of the accumulated values for a given month, which is then transformed to a standardized normal variate (Z-score) with a mean of zero and a standard deviation of one. This process is repeated for all the months (McKee et al., 1993). Standardization enables fair comparison of drought conditions across the diverse study area. Standardized values below (above) zero indicate dry (wet) conditions. The Pearson correlation between monthly SSI (at a one-month timescale) and SPI across various timescales (TS = 1 to 12) is calculated for each month. The SPI timescale that shows the highest correlation with SSI indicates the drought propagation duration for that month (Barker et al., 2016). Details of the precipitation and soil moisture data used to compute these indices are provided in Section 3.

**2.1.2 Drought propagation probability**

The propagation of meteorological to agricultural drought, as shown in Fig. 1a, is assessed by the probability of agricultural drought occurrence (SSI ≤ -0.5) given the presence of pre-existing accumulated meteorological droughts over the propagation duration ($SPI_{TS}$ ≤ -0.5) (Long et al., 2024; Xu et al., 2021):

$$P(SSI \leq -0.5 \mid SPI_{TS} \leq -0.5) = \frac{C(G(-0.5), \ F(-0.5))}{F(-0.5)} \tag{1}$$

where G and F represent the marginal distributions of the random variables SSI and SPI (for the chosen timescale), respectively, while C denotes the bivariate copula function fitted to model the dependence between the two variables. The optimal copula function is selected from Clayton, Frank, Gaussian, Gumbel, Joe, Student t, BB1, BB6, BB7, and BB8 families based on maximum log-likelihood criterion (Xu et al., 2023). The drought propagation duration and conditional probability (Eq. 1) used to define propagation are computed for each grid and for every month across the study area.

**2.1.3 Factors influencing soil moisture and propagation using Random Forest (RF) models**

Random Forest (RF) models have recently been employed to identify the primary drivers of drought propagation (Dai et al., 2022; Li et al., 2023). These studies make use of interpretable outputs, such as the variable importance feature, to rank the influential factors. Since meteorological droughts contribute to antecedent soil-moisture (SM) deficits, precipitation (Pr) is one of the key factors included in the RF model. Another important climatic factor is evapotranspiration, which, in turn, is influenced by temperature (T), humidity (H), vegetation cover (VC), solar radiation (SR), and wind (W) (Guo et al., 2017). These climatic variables serve as predictors, with soil moisture as the predictand in the RF models ($f_1$) (Breiman, 2001):

$$SM = f_1(T, Pr, H, VC, SR, W, TS) \tag{2}$$

To account for the memory effect of soil moisture, the predictors are standardized using timescales (TS) ranging from 1 to 12 months as one of the predictor variables (Muthuvel et al., 2023). The RF models are trained and tested using an 80:20 random data split for each grid point in the study area. Their performance is evaluated based on the coefficient of determination ($R^2$)

and Root Mean Square Error (RMSE) values, calculated from the predicted and observed values in the testing dataset. Due to the computational intensity involved in training 5364 (1788 grids X 3 timescales), a fixed set of hyperparameters are used. The three fixed hyperparameters are the number of trees (ntree = 500) following Dai et al. (2022), the number of variables tried at each split (mtry = 5), and the minimum size of terminal nodes (nodesize = 5). These temporal RF models predicting soil moisture do not explicitly relate to soil moisture deficit. To address this, and to complement the grid-specific temporal RF models, spatial RF models that directly predict drought propagation probabilities (CP = P(SSI<-0.5|SPI<-0.5) in Eq. 1) for each timeframe (three RF models in total) are also modelled.

Hu et al. (2024) developed spatial RF models to assess the relative importance of variables influencing drought propagation. In the present study, these models are considered as spatial because each grid cell is associated with a single propagation probability value (the predictand), forming one row in the training dataset. The predictors include elevation, climate zones, season (represented by month), and the monthly means of various climate variables (i.e. temperature: $T_{Mean}$, precipitation: $Pr_{Mean}$, humidity: $H_{Mean}$, vegetation cover: $VC_{Mean}$, solar radiation at 100 m: $SR_{Mean}$, and wind speed: $W_{Mean}$). The spatial RF model ($f_2$) predicting propagation probability (CP) is expressed as follows:

$$CP = f_2(T_{Mean}, Pr_{Mean}, H_{Mean}, VC_{Mean}, SR_{Mean}, W_{Mean}, Elevation, Climate\ Zone, Month) \qquad (3)$$

Since propagation probability is computed over an entire timeframe, the climate variables are aggregated as monthly means for each timeframe. The spatial RF models capture the influence of stationary predictors (e.g., elevation and climate zone), which temporal models cannot. Climate zones are defined based on the Köppen-Geiger climate classification (Kottek et al., 2006). In contrast, the temporal RF models (5,364 in total: 1,788 grids x 3 timeframes) capture temporal climatic variations, with each month's data forming a row in the training dataset for a given grid-specific model. These temporal variations are aggregated in the spatial RF models. To predict drought propagation probabilities across timeframes using spatial RF models (three models, one per timeframe), extensive grid-based hyperparameter tuning with cross-validation is conducted. The dataset consists of 1,788 spatial samples, randomly divided into training and testing sets. Five-fold cross-validation is performed by partitioning the training set into five equal subsets. The model is trained on four folds and validated on the fifth, repeating the process five times so that each fold serves once as the validation set. The average $R^2$ value across the folds is used to guide the hyperparameters tuning. Hyperparameters are chosen from all combinations of ntree = {100, 500, 1000, 1500}, mtry = {2, 5, 7, 9}, and nodesize = {1, 5, 10, 15, 20} based on cross-validation $R^2$ values. Final predictions are made on the testing set using the optimal hyperparameters, and model performance is evaluated using $R^2$ (Coefficient of determination), BIAS (Mean Bias Error), MAE (Mean Absolute Error), MSE (Mean Squared Error), and RMSE (Root Mean Squared Error) values. The most influential variables in prediction accuracy are ranked for both RF models based on %IncMSE values (percentage increase in mean squared error). Variables with higher %IncMSE are those most influential in determining soil moisture (Eq. 2) and drought propagation (Eq. 3).

### 2.1.4 Relationship between rainfall deficit and agricultural droughts

To check if an agricultural drought is associated to a prior rainfall deficit (below normal rainfall denoted by SPI<0), the reverse of propagation probability (Eq. 1) is derived as follows:

$$P(SPI_{TS} \leq 0 | SSI \leq -0.5) = \frac{C(F(0), \; G(-0.5))}{G(-0.5)} \tag{4}$$

where G and F represent the marginal distributions of the random variables SSI and SPI (for the chosen timescale TS), respectively, while C denotes the bivariate copula function fitted to model the dependence between the two variables. This reverse propagation probability reflects the extent to which an agricultural drought is linked to a prior meteorological drought.

## 2.2 Drought persistence

While monsoonal droughts are a key focus, given that these months account for the majority of annual rainfall, droughts during the pre- and post-monsoon periods can also severely impact crop growth. Yang et al. (2021) highlight that soil moisture deficits prior to planting can impair seedling root development, significantly affecting crop yields. Thus, droughts in pre-monsoon that coincide with land preparation and early sowing stages must also be analysed. Moreover, residual soil moisture from the monsoon is crucial for winter (or post-monsoon) crop growth. Using joint probabilities, eight different scenarios of inter-seasonal droughts can be measured from the three seasons, as shown in Fig. 1b. The worst-case scenario is the persistent all-season droughts. To this end, agricultural drought indices during the end months of Pre-Monsoon, Monsoon, and Post-Monsoon seasons are used as random variables. For regions above the equator, SSI values during May, September, and December are considered to represent the three respective seasons (Muetzelfeldt et al., 2021; Zhou et al., 2011). For regions in the equator and below the equator in SEA, the months of Sep, Jan, and May are used to study the inter-seasonal relationship since some of the grids receive rainfall during both Summer and Winter Monsoons (Van Noordwijk et al., 2017; Singh and Qin, 2020). Fang et al. (2019) used bivariate joint distributions considering indices during wet and dry seasons. Extending it to three seasonal drought variables, the probability of persistent agricultural droughts (defined by SSI) during the pre-monsoon, monsoon, and post-monsoon months in a given water year are estimated as:

$$P(SSI_{Pre-Monsoon} \leq i, SSI_{Monsoon} \leq i, SSI_{Post-Monsoon} \leq i) = C(F(i), G(i), H(i)) \tag{5}$$

where F, G, and H are the marginal distributions of the three seasonal SSI values, while i refers to the drought severity thresholds. Shah and Mishra (2020) suggested using Gaussian copula for multivariate analysis involving more than two drought index values. Hence, the present study uses Gaussian copula (C) to represent the dependence among the three random variables of seasonal droughts. In terms of return periods ($T_{Persistent\ Droughts}$), drought persistence for a given severity threshold (i) is derived as (Fang et al., 2019):

$$T_{Persistent\ Droughts} = 1 / P(SSI_{Pre-Monsoon} \leq i, SSI_{Monsoon} \leq i, SSI_{Post-Monsoon} \leq i) \tag{6}$$

Apart from the worst-case scenario of persistent all-season droughts, there are other scenarios, such as two-season droughts (only two of the three seasons are drought-affected in a water year), isolated single-season droughts (only one of the three seasons is drought-affected in a water year), and no-drought (all three seasons in a water year have above normal SSI)

scenarios. All eight scenarios will be computed using Gaussian copula (C) based on trivariate probabilities (similar to Eq. 5) described in Table S1.

## 2.3 Spatial concurrence

One of the most commonly used drought attributes is the spatial coverage of drought, measured as the percentage of grids experiencing drought (SSI < i) within a given region during a specific month. Following the approach by Yu et al. (2023), which uses copula-based conditional probability to analyse spatial concurrence (Fig. 1c), if $Reg_2$ and $Reg_1$ represent the spatial agricultural drought coverages (SSI < i) of two regions, the drought concurrence between them is defined as:

$$P(Reg_2 \geq k \mid Reg_1 \geq k) = \frac{1 - F(k) - G(k) + C(F(k), G(k))}{1 - F(k)} \qquad (7)$$

where k is the threshold of spatial drought coverage, and F and G are the marginal distributions of $Reg_1$ and $Reg_2$ respectively. The bivariate return period to estimate monsoonal concurrent droughts ($T_{CD}$) are derived as (Muthuvel and Mahesha, 2021):

$$T_{CD} = 1 / P(Reg_2 \geq k, Reg_1 \geq k) = 1 / (1 - F(k) - G(k) + C(F(k), G(k))) \qquad (8)$$

After fitting appropriate bivariate copula C (Similar to Eq. 1), conditional probability curves (Eq. 7) and return period contours (Eq. 8) for various drought coverage thresholds (k) will be plotted. Figure 1c shows the joint (upper right shaded domain) and

conditional (upper right and right shaded domains) probability domains between the two regions exceeding the drought coverage threshold (k) used to derive Eq. (7) and (8). While empirical marginal distributions are used in equations 1, 4, and 5, appropriate theoretical marginal distributions, selected from Generalized Extreme Value, Gamma, Gumbel, Lognormal, and Pearson Type III distributions based on the lowest Akaike information criterion (AIC) values, are fitted in Eq. (7) and (8) to produce smooth drought frequency curves (Datta and Reddy, 2023). The entire analysis is conducted in the R programming

language (v4.3.3; R Core Team 2024), utilizing packages such as lmomco (Asquith, 2024) for marginal distribution fitting, rvinecopulib (Nagler and Vatter, 2023) for copula functions, rnaturalearth (Massicotte and South, 2023) for mapping, and ggplot2 (Wickham, 2016) for figure plotting.

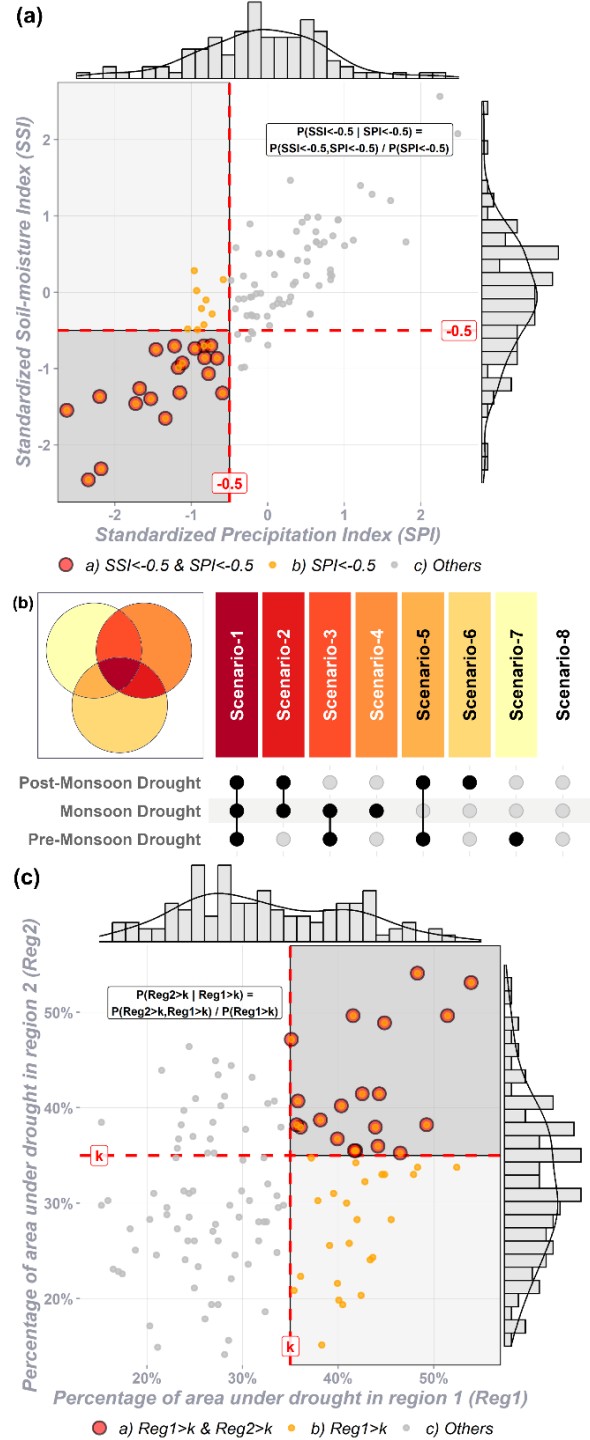

**Figure 1: (a) Domain of conditional probability defining the propagation from meteorological to agricultural drought, (b) Venn diagram of all possible inter-seasonal drought scenarios, and (c) domain of conditional and joint exceedance probability for studying drought spatial concurrence.**

## 3 Study Area and Data

The study area of monsoon-dominant Asia (Fig. 2a and Fig. 2b) comprises countries in the domain of Tibet (TIB), South Asia (SAS), East Asia (EAS), and Southeast Asia (SEA). The grid points at 1º x 1º spatial resolution within the administrative boundaries of 21 countries are used in the study. The observed monthly precipitation and soil moisture data between 1975 and 2014 from GLDAS (Global Land Data Assimilation System, version 2.0) are used for reference (Rodell et al., 2004). The study uses the worst-case emission scenario of SSP5-8.5 (Shared Socioeconomic Pathway) from Coupled Model Intercomparison Project phase 6 (CMIP6) for future projections. Multi-Model Ensembles (MMEs) of monthly precipitation (pr) and soil moisture (mrso) are calculated using Bayesian Model Averaging (BMA) based on simulated historical climate and observed data using R library BMA (Raftery et al., 2024). While MMEs using simple averages with equal weights for all models have proven effective in studying droughts (Feng et al., 2023), the potential over- or underestimation of variables by individual models can be mitigated by assigning appropriate weights through BMA (Zhai et al., 2020; Muthuvel et al., 2023). A detailed explanation of the MMEs based on the Bayesian averaging method is provided in Tian et al. (2025). A comparison of the spatial monthly distributions between the MMEs and observed data reveals substantial similarity for both rainfall (Fig. 2c) and soil moisture (Fig. S1 in the Supplementary Materials). Additionally, the MME outperforms the eight individual models (BCC-CSM2-MR, CanESM5, CNRM-CM6-1, CNRM-ESM2-1, GFDL-ESM4, IPSL-CM6A-LR, MIROC-ES2L, and MRI-ESM2-0) (Fig. S2 and Fig. S3) and will therefore be used to compute the drought indices. Furthermore, monthly plots of the mean differences between GCM data (Multi-Model Mean using Bayesian Model Averaging) and GLDAS (Observed data) show that the median monthly precipitation difference is within ±10 mm for most months across all regions (Fig. S4a). Similarly, the median soil moisture differences remain within ±20 mm across all regions and months, substantially smaller compared to the observed median monthly soil moisture values, which are, around 500 mm (Fig. S4b). Notably, the differences in cumulative drought severities between the observed data and the historical MME indicate that most regions exhibit differences within ±10% for both meteorological (except for certain grids in Western SAS and TIB regions with ±20% difference) and agricultural droughts (Fig. S5). The individual GCMs are interpolated and regridded to a uniform spatial resolution of 1º x 1º. These selected climate models also provide access to additional monthly variables, including temperature (tas), humidity (huss), vegetation cover (lai), radiation (rsds), and wind (sfcWind), which will be used in the RF models (Equations 2 and 3). The climate models are freely available and managed by the World Climate Research Program (WCRP, 2024).

GLDAS datasets are widely employed in drought research and have been shown to effectively capture major historical drought events. For instance, we plotted the percentage of areas affected by meteorological and agricultural droughts using GLDAS data (Fig. S6), which clearly reflects significant events such as the 2009 drought (Barriopedro et al., 2012). Additionally, Gupta and Karthikeyan (2024) reported good agreement in meteorological drought characteristics across MSWEP (Multi-Source Weighted-Ensemble Precipitation), CHIRPS (Climate Hazards group InfraRed Precipitation with Stations data), and GLDAS, supporting the reliability of GLDAS. While MSWEP and GLEAM (Global Land Evaporation

Amsterdam Model) are excellent datasets for drought propagation studies, existing literature suggests that the GLDAS performs comparably well. Furthermore, the three timeframes used in the study, historical (1975-2014), near-future (2021-2060, and far-future (2061-2100), were selected to ensure equal-length epochs of 40 years for consistent comparison of drought propagation, persistence, and spatial concurrence. However, MSWEP and GLEAM begin only in 1979 and 1980, respectively, making them unsuitable for this time range. Given these considerations, GLDAS is an appropriate choice for the present study.

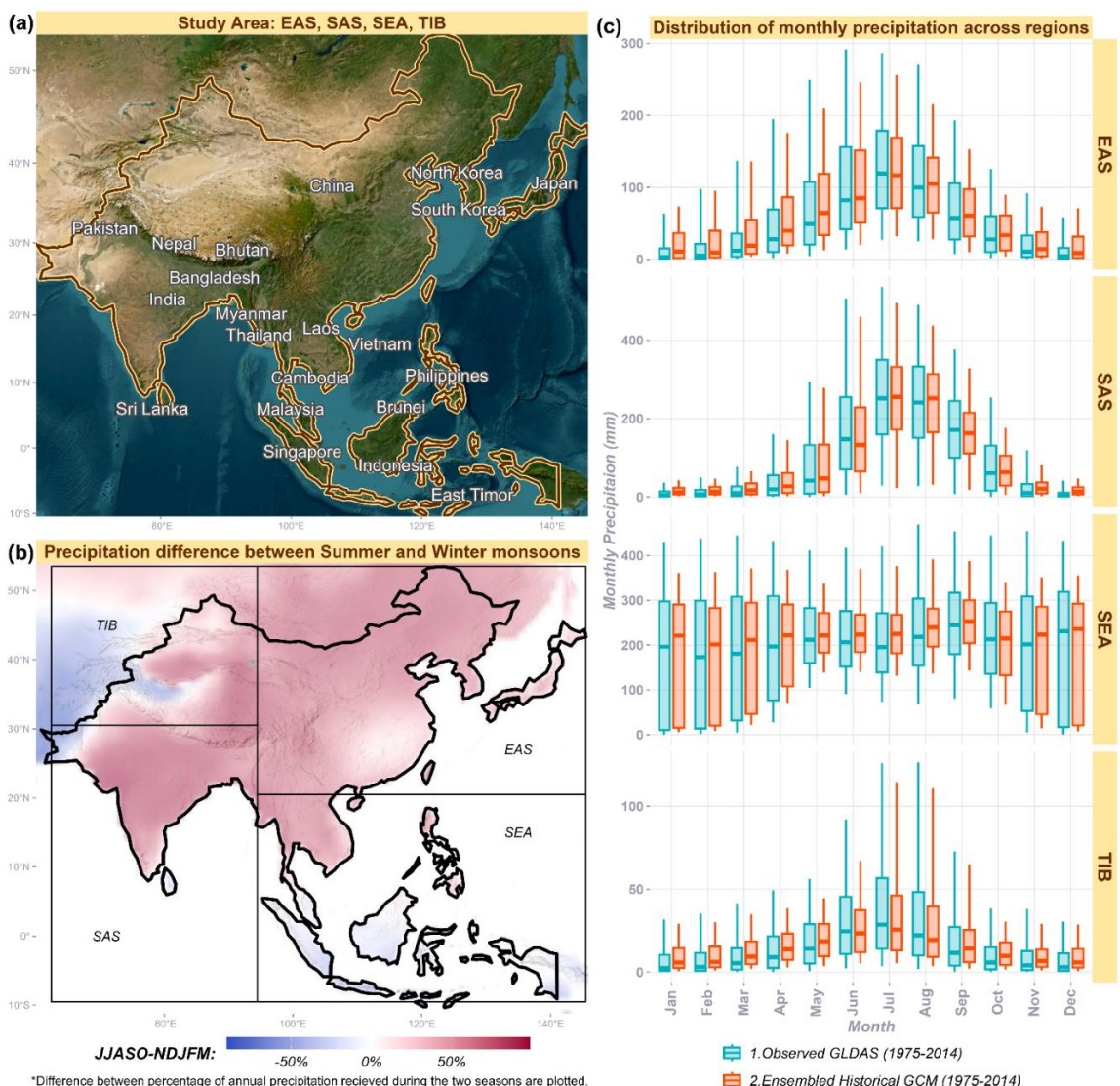

**Figure 2: (a) Study area of monsoon-dominant Asia with its boundary, (b) percentage difference in precipitation between Summer (June-October) and Winter (November-March) monsoons, regional divisions are based on thematic maps from works such as Giorgi and Bi (2009) and Sillmann et al. (2013), and (c) monthly comparison of observed and historical GCM precipitation data across four regions.**

# 4 Results and discussion

### 4.1.1 Drought propagation duration

Initially, propagation durations based on the timescales of SPI (TS between 1 and 12) corresponding to the maximum correlation with monthly SSI (TS = 1) are estimated for each month across the entire study area, with their distributions presented in Fig. 3a. Unlike earlier studies (Guo et al., 2020) which considered all months together to estimate drought propagation duration, calculating propagation duration for each month is crucial in monsoon-dominant regions with pronounced seasonal rainfall disparities. During the historical period, approximately 32% of the study area exhibited propagation durations of three months or less in June. The percentage of the area experiencing such short-term rapid propagation durations (TS = 1 to 3) steadily increase to 55% by October before progressively declining to 23% by April. Conversely, the percentage of the area with prolonged propagation duration (TS > 6 months) increases steadily from 25% in October to 55% in May, followed by sharp decline during the monsoon months (June to October). This patten aligns with the seasonal variability associated with the onset and withdrawal of the monsoon in the region. Soil moisture replenishes with the onset of the monsoon, as indicated by shorter timescales from June to October. Most areas in the study region receive substantial precipitation during this period, and soil moisture is heavily dependent on it.

Owing to the sensitivity of soil-moisture to monsoon precipitations, prolonged propagation durations are observed in the post-monsoon months. Notably, parts of arid SAS and TIB exhibit propagation durations up to 12 months during May in the historical timeframe, as shown in Fig. 3b. From the maps of propagation durations (Fig. S7), regions in SEA regions, such as Indonesia and the Malay Peninsula, show comparatively shorter propagation durations despite differences in seasonality. Unlike the temporal skewness of rainfall distribution during the monsoon in the study area, rainfall in these regions is more evenly distributed across most months of the water year. The percentage distribution of propagation durations in the near (2021-2060) and far-future (2061-2100) remains largely similar to that of the historical timeframe (1975-2014), aligning with the monsoon progression. Similar findings of minimal shifts in propagation durations under climate change have been reported in studies from South China (Zhou et al., 2023, 2021), albeit for propagation from meteorological to hydrological droughts.

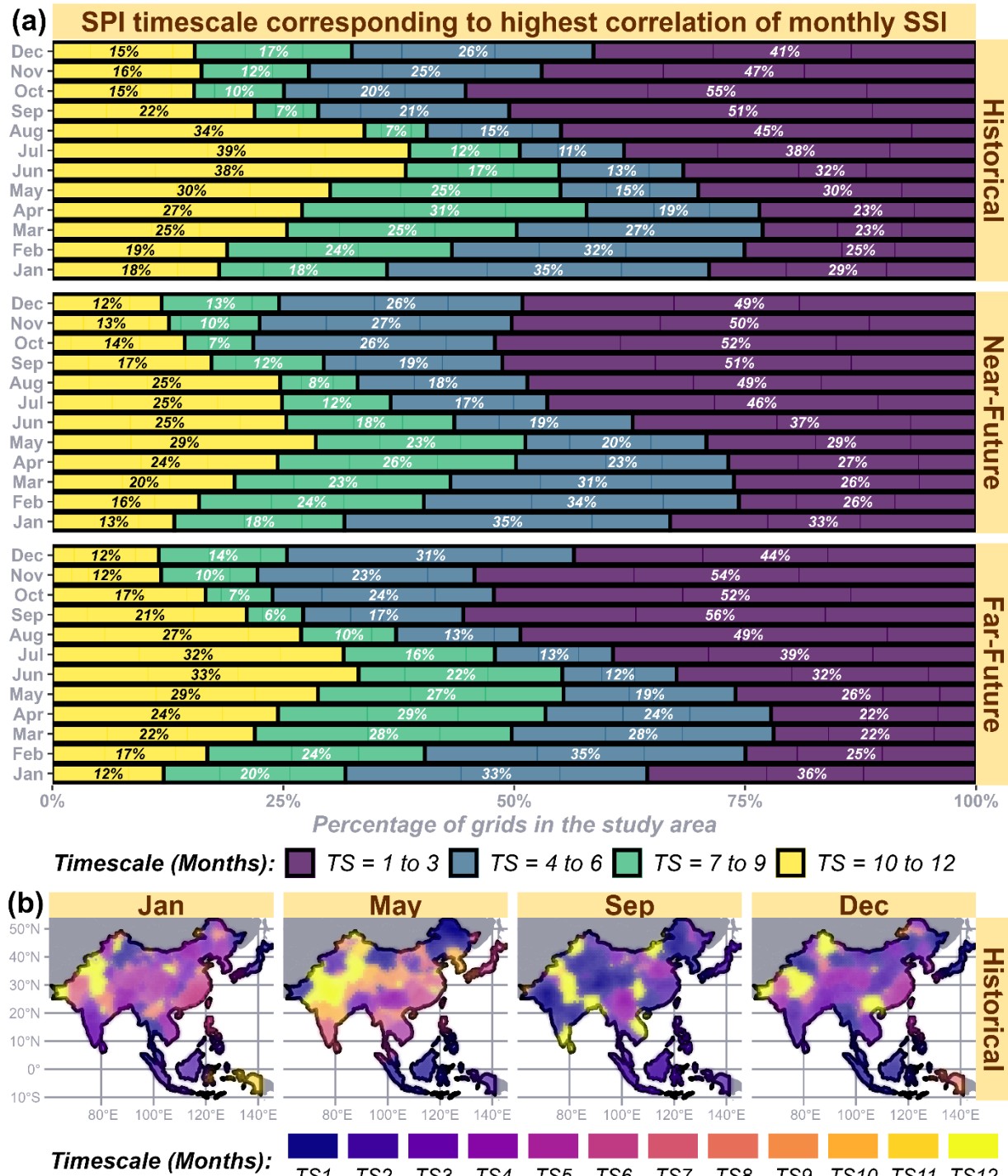

Figure 3: Propagation durations (in timescale, TS) shown as (a) percentages across the study area for each month in different timeframes and (b) spatial maps for different seasons in the historical timeframe.

### 4.1.2 Drought propagation probability

Conditional probabilities (CP), which defines the propagation of agricultural drought from meteorological droughts for each month, are calculated based on the previously computed propagation durations (SPI timescales, Fig. 4). Comparing the median CP values, half of the grids in EAS exhibit propagation probabilities below 0.25 in the historical timeframe across all seasons (Fig. 4a). In the near- and far-future timeframes, the median CP values increase to approximately 0.41 and 0.52 across seasons, respectively. In SAS, the historical median CP reaches its highest value in September (0.33) at the end of the monsoon season, followed by December (0.27), January (0.23), and May (0.2) towards the region's water year end. Similar to EAS, SAS shows higher median CP values in the near and far-future timeframes compared to the historical period. However, unlike the historical timeframe, future propagation probabilities in SAS during non-monsoon months (median CPs of 0.58, 0.61, and 0.64 for December, January, and May, respectively, in the near-future) exceed those of monsoon months (median CP of 0.44 for September in the near-future). In SEA, the distribution of CP values remains largely consistent between the historical and near-future timeframes across all seasons, indicating that drought propagation probabilities remain stable. However, in the far-future timeframe, median CP values drop significantly compared to the other two periods. For instance, the historical median CP of about 0.42 across all months declines sharply to 0.25, 0.34, 0.32, and 0.24 during January, May, September, and December, respectively, in the far-future. The reduction in CP values is even more pronounced in TIB, where the historical median CP of approximately 0.50 decreases to 0.15 and 0.12 in the near- and far-future timeframes, respectively. This decline suggests that meteorological droughts are less likely to evolve into agricultural droughts in the future compared to the historical period, which is a positive indication for grids in TIB and SEA. Despite minimal shifts in the propagation durations estimated earlier, significant changes in propagation probabilities are expected across the study area in the future.

The difference in CP values during the near and far-future compared to historical timeframes are spatially depicted in Fig. 4b. Regions with increased propagation probability in the far-future across seasons, relative to the historical period, include most of SAS (excluding Peninsular and Western India) and Eastern China, making these areas more vulnerable to meteorological droughts transitioning into agricultural droughts. These regions have also been identified as prone to compound dry hot events (the coexistence of meteorological droughts and above-normal hot events) under future climate change scenarios (Feng et al., 2025; Prabhakar et al., 2023). Rising temperatures associated with climate change are likely to exacerbate this risk by depleting soil moisture and accelerating the drought propagation process. In contrast, regions such as the arid Xinjiang province in northwestern China (TIB domain) and Mainland Southeast Asia (except during May) are expected to experience reduced drought propagation in far-future compared to the historical timeframe. Previous studies (Try and Qin, 2024; Wang et al., 2022) project more frequent extreme precipitation events in these areas under the SSP5-8.5 scenario, which could mitigate drought propagation.

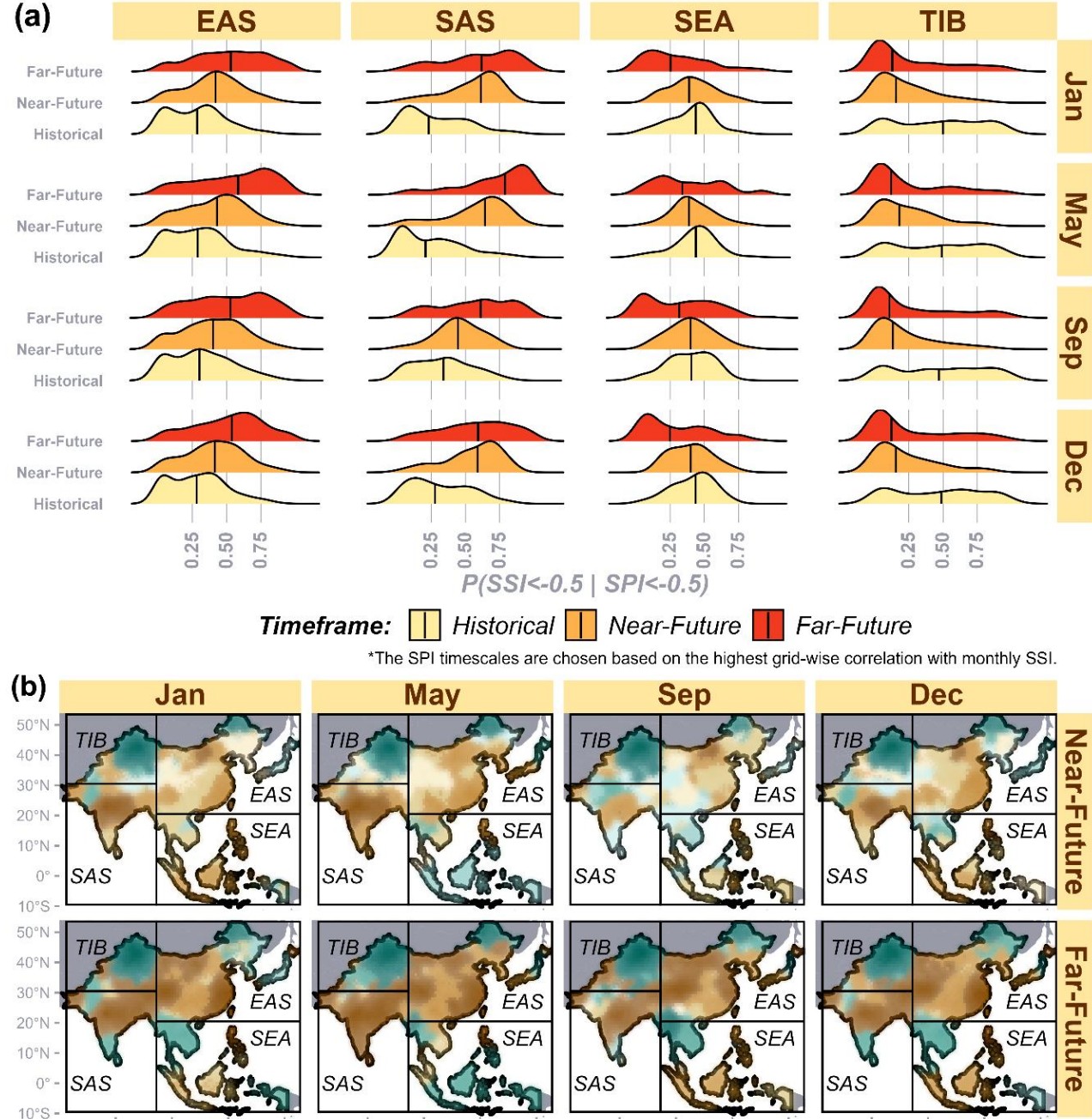

**Figure 4: (a)** Comparison of propagation probabilities across four regions, seasons, and timeframes, and **(b)** spatial maps showing the differences in propagation probabilities between future and historical timeframes.

### 4.1.3 Factors affecting soil moisture and propagation

Several climatic factors contribute to soil moisture depletion and exacerbate the drought propagation process. Key drivers identified by previous studies (Wang et al., 2023) include precipitation, air temperature, specific humidity, solar radiation, near-surface wind speed, and leaf area index (vegetation), all of which are used as predictors in RF models to estimate soil moisture. To account for their lag effects on monthly SSI, standardized indices of these variables at timescales ranging from 1 to 12 months are incorporated into the model. An RF model is developed for each grid, and its performances is evaluated using the coefficient of determination ($R^2$). Overall, RF models perform better in the future timeframes, with most achieving $R^2$ values above 0.75 in the far-future and above 0.45 in the near-future, compared to lower $R^2$ values in the historical period (Fig. 5a). RMSE (standardized soil moisture) values are much lower (less than 0.1) in the TIB region across timeframes (Fig. S8). RMSE values are higher (around 0.7) in SAS and southern EAS in the historical and far-future timeframes, respectively. The distribution of variable importance in the historical timeframe reveals similar quantile values for temperature, humidity, and radiation (Fig. 5b), although temperature tends to have higher quantiles. The predictor with the highest variable importance is identified as the key driver. As shown in Fig. 5c, in the historical timeframe, humidity (32% of the study area) and vegetation cover (25% of the study area) are the primary drivers of agricultural droughts. However, in future timeframes, temperature emerges as the dominant driver in nearly half of the study area (see Fig. S9 in the Supplementary Materials). This finding, highlighting the increasing influence of temperature on soil moisture, aligns with the projected rise in compound drought and heatwave events reported by Tripathy et al. (2023). Despite low $R^2$ values in some cases, the variable with the highest feature importance still demonstrates a relatively stronger influence compared to the other climate predictors. Apart from the climatic variables, other grid-specific predictors such as elevation (Zhang et al., 2024) and climate characteristics (Zhang et al., 2021), which play a crucial role, are not included in these RF models. Since these variables are temporally constant for a given grid, they do not affect the time series prediction. Consequently, at certain grid points with low $R^2$ values, factors beyond the selected climate variables may influence soil moisture.

To complement these temporal RF models, spatial RF models incorporating grid-specific predictors, such as elevation and climate classification, along with aggregated climate variables, are used to predict propagation probability. The cross-validated $R^2$ values are computed for all combinations of hyperparameters. The hyperparameter mtry (the number of variables tried at each split) proves to be the most sensitive, with higher values (9 out of 10 variables) performing well (yielding higher $R^2$ values across validation folds) compared to ntree (number of trees) and nodesize (minimum size of terminal nodes) (Fig. 6a). A combination of higher mtry and lower nodesize improves the predictive skills of the RF models, regardless of the ntree values. The hyperparameter combination of ntree = 1000, mtry = 9, and nodesize = 1 yields the best RF models, with $R^2$ values of 0.78 and 0.64 in the cross-validation sets of historical and far-future timeframes, respectively. The RF model corresponding to the near-future timeframe performs the best with hyperparameter values of ntree = 1500, mtry = 9, and nodesize = 1, yielding $R^2$ values of 0.78 in the validation phase. The three RF models with these chosen sets of hyperparameters are then used to predict propagation probabilities from the remaining 20% testing sample (Fig. 6b). Evaluating the predictions against the

observed samples in the testing phase shows that the historical (RMSE = 0.12, BIAS = -0.002, $R^2$ = 0.79) and near-future (RMSE = 0.11, BIAS = -0.003, $R^2$ = 0.79) RF models perform better than the far-future (RMSE = 0.18, BIAS = -0.001, $R^2$ = 0.67) RF model. Monthly mean soil moisture is the most influential variable, followed by elevation and climate characteristics in the historical timeframe (Fig. 6c). In both future timeframes, climate characteristics (Fig. S10 in the Supplementary Materials) will become influential factors in determining the propagation probability. Elevation and monthly mean soil moisture are the second and third most influential factors in the near-future timeframe, with their influence reversed in the far-future. Vegetation cover and monthly mean wind speed are consistently ranked fourth and fifth across timeframes. These results affirm that climate characteristics and elevation, which are key spatial drivers of drought propagation, are consistent with findings from previous studies (Sadhwani and Eldho, 2024; Shi et al., 2022; Zhang et al., 2022). Shi et al. (2022) found that the propagation duration from meteorological to hydrological droughts varies globally according to climatic zones. Similarly, the probability of meteorological to agricultural droughts depends primarily on a combination of climate characteristics, elevation, and mean soil moisture. Besides mean soil moisture directly affecting propagation probability, climate characteristics summarize multiple climate features (Peel et al., 2007), while elevation slope controls runoff and infiltration (Zhang et al., 2022). In the temporal RF models predicting monthly soil moisture, temporally dynamic temperature plays a key role. Meanwhile, in the spatial RF model predicting propagation probability, the mean temperature is aggregated over 40-year windows, which reduces its temporal variability and predictive utility. Instead, spatially varying factors such as climate characteristics and elevation dominate the prediction of propagation probability.

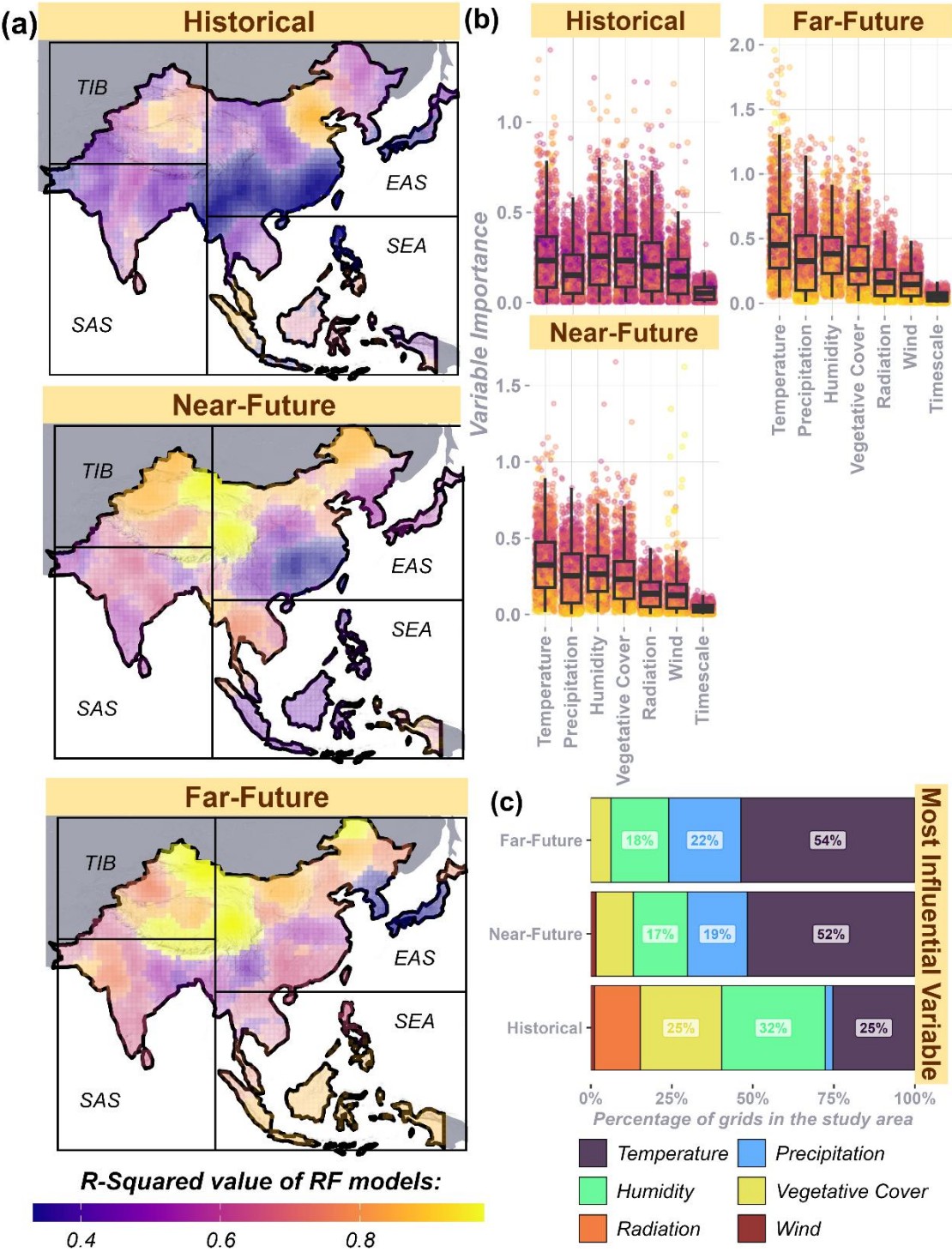

**Figure 5: (a)** Performance evaluation (R² values) of Random Forest (RF) models for each grid across timeframes, **(b)** variable importance values of predictors used in the RF models, and **(c)** most important variable (highest importance value in an RF model) in each timeframe across the study area.

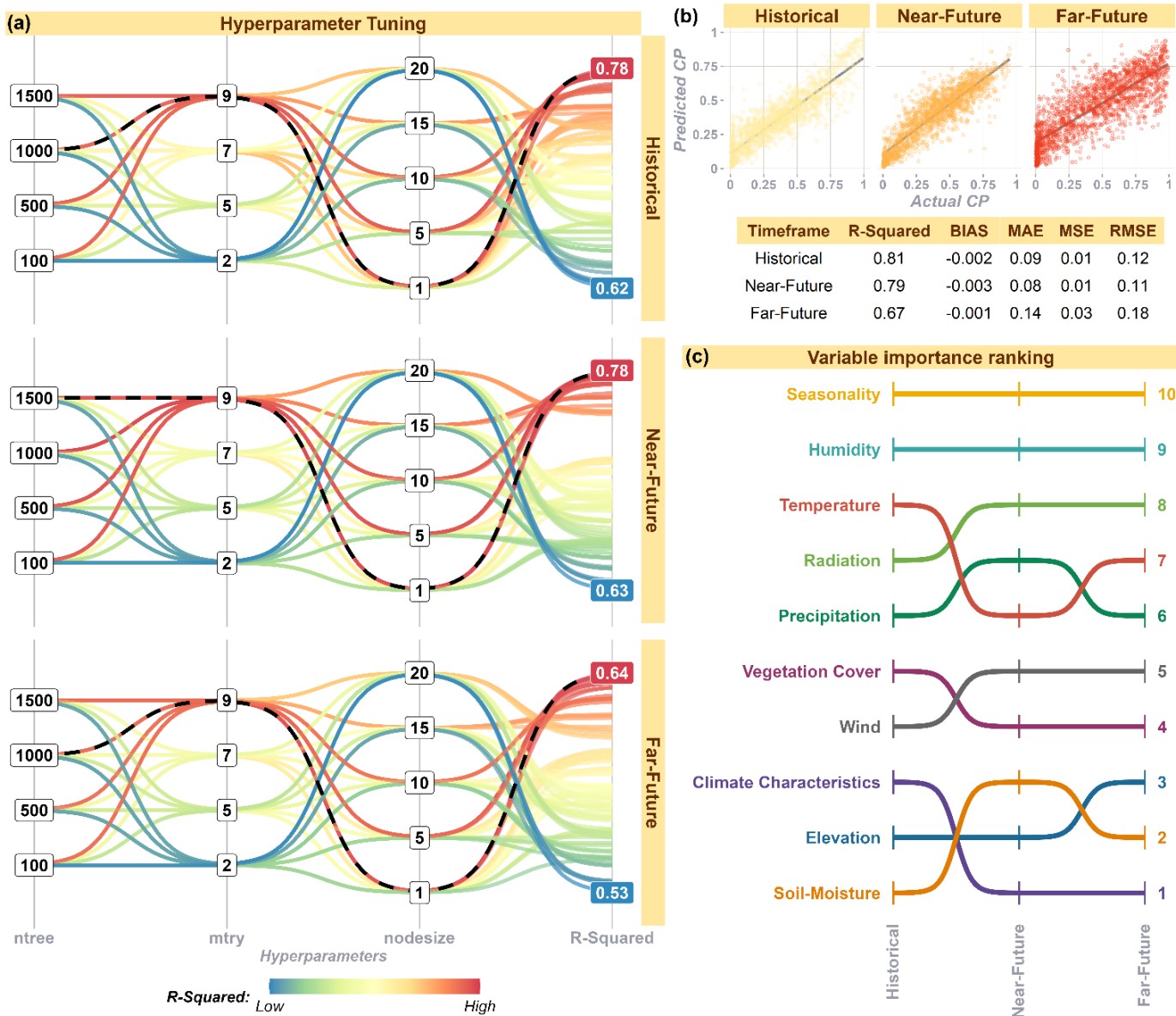

**Figure 6:** (a) Grid-based hyperparameter tuning for the spatial RF models using cross-validation (The chosen sets of hyperparameters are connected by dotted lines), (b) performance evaluation of the RF models designed using chosen sets of hyperparameters in the testing dataset, and (c) most important variable (highest importance value in an RF model) in each timeframe.

### 4.1.4 Relationship between rainfall deficit and agricultural droughts

This reverse propagation probability, (P(SPI<0|SSI<-0.5)), reflects the extent to which an agricultural drought is linked to a prior meteorological drought (Fig. 7). The density plots show a decline in reverse propagation probability values in both the near- and far-future scenarios relative to the historical period across all regions, with the most significant decrease observed in EAS in the far-future timeframe. This suggests that the number of agricultural drought events not directly attributable to meteorological droughts is expected to rise. Simultaneously, the forward propagation probability (P(SSI<0|SPI<-0.5))

indicates that meteorological droughts are increasingly driving agricultural droughts in SAS and EAS in the future. Thus, while meteorological-driven agricultural droughts are projected to increase, so too are those unrelated to rainfall deficits. Random forest models used to predict soil moisture further reveal that temperature becomes the dominant driver of soil moisture across more than 50% of the study area in future scenarios, compared to about 25% in the historical period (Fig. 5(c)). This supports the observed increase in non-rainfall-related agricultural droughts. Under the SSP5-8.5 scenario, significant temperature

increases are expected towards the end of the century (Qiao et al. 2023). This warming could intensity soil moisture deficits, leading to a shift from meteorological-driven to temperature-driven agricultural droughts. Additionally, above-average rainfall (SPI>0) may occur in the form of short-term, intense storms, which may not adequately replenish soil moisture under higher temperatures, which is a situation exacerbated by climate change. In EAS, and SAS, although meteorological-driven agricultural droughts are projected to rise in the far-future, they are likely to be more severe due to the increasing influence of

temperature, potentially resulting in more frequent compound drought-heatwave events. These findings support the methodological decision to use SSI directly to assess drought persistence and concurrence, rather than relying solely on SPI. This ensures that agricultural droughts driven by rainfall deficits and non-rainfall-related droughts are considered for analysing persistence and cross-regional concurrence.

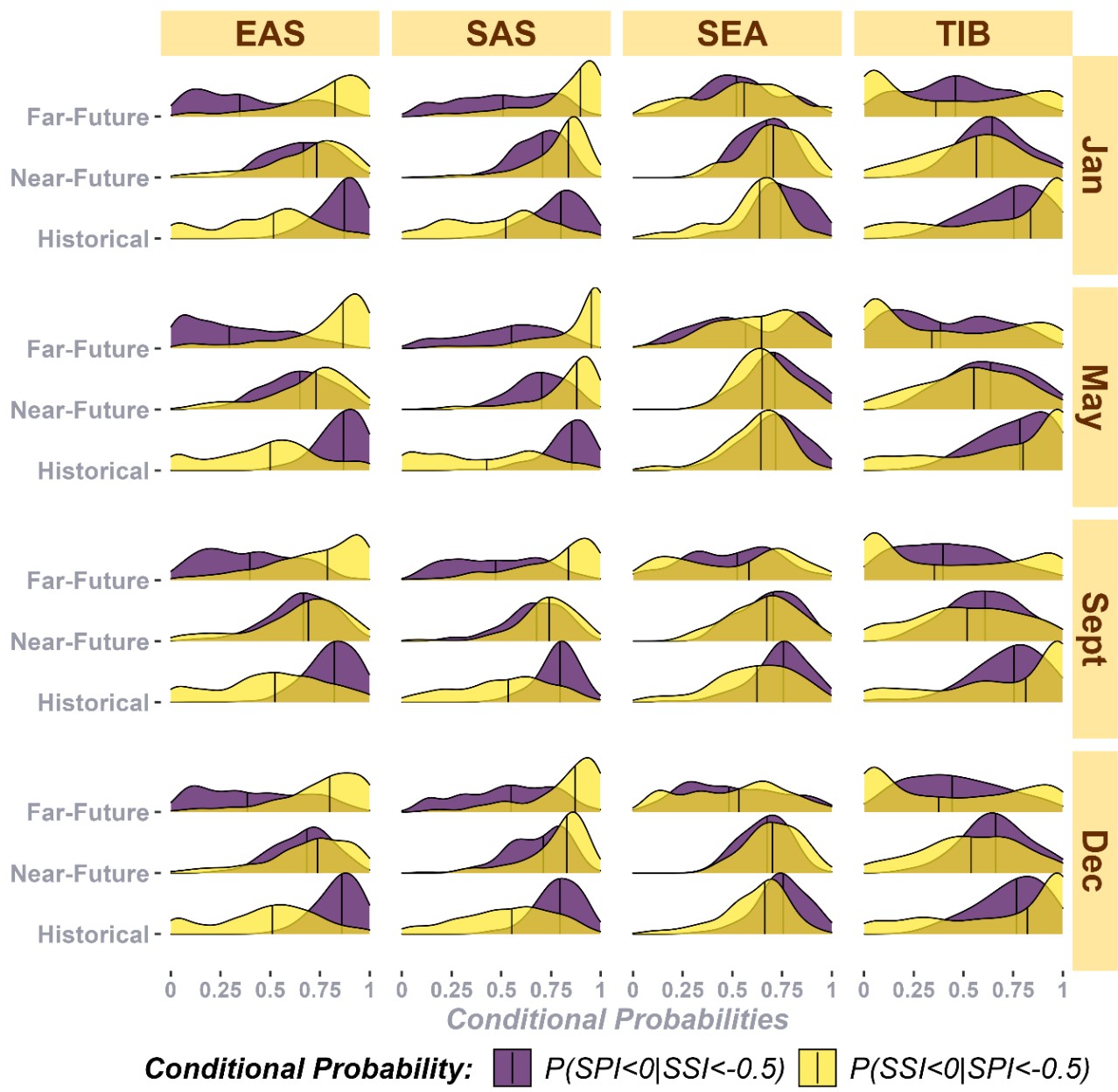

**Figure 7: Reverse propagation probability (P(SPI<0|SSI<-.5)) indicates if an agricultural drought is linked to a prior meteorological drought.**

## 4.2 All-season persistent droughts

Having identified regions sensitive to the propagation of meteorological to agricultural droughts, it is important to also focus on the persistence of agricultural droughts across seasons. The joint probability (JP) of agricultural droughts

(monthly SSI < 0) during the pre-monsoon, monsoon, and post-monsoon months in a given water year is used to define persistence (Fig. 8a). Among the eight possible combinations of seasonal droughts, the probability of all-season droughts and its complementary scenario of all wet seasons are higher than those isolated seasonal droughts. In the historical timeframe, half of the study area has JP values more than 0.17 for all season droughts, which increases to about 0.25 in both the near- and far-future timeframes. In contrast to the increase in all-season droughts, the median JP for all-season wetness decreases from 0.25 in the historical timeframe to 0.2 and 0.18 in the near- and far-future, respectively. When plotting the difference in JP values between the future timeframes and the historical timeframe, regions identified as at higher risk of all-season droughts in the near-future include central India (in SAS), regions in EAS, and the Philippines (Fig. 8b). The differences in JP values are more pronounced in the far-future, indicating an amplified risk of persistent all-season droughts. Liang et al. (2023) reported that CMIP6 models predict an increase in both extreme and mean precipitation values in northwestern China (in the TIB domain) in the future, compared to the historical values, while a decrease in precipitation is projected for the southwestern region (in the EAS domain). The current results also reflect this pattern, with an increase (and decrease) in future JP values in these regions, albeit from a drought perspective.

Regarding return periods of all-season persistent droughts (Fig. 9), northwestern China (in the TIB domain) exhibits a return period of less than 2 years for SSI < 0 in the historical timeframe, which increases drastically to over 50 years in the far-future (Fig. 9a). When comparing across timeframes, severe persistent all-season droughts (SSI < -0.8 across seasons) are historically rare, occurring only once in more than 50 years in approximately 80% of the study area (Fig. 9b). However, such events are projected to become more frequent in the future. Specifically, only about 12% of the study area experiences frequent severe persistent all-season droughts (return periods of less than 20 years) in the historical timeframe, but this increases to 23% and 39% in the near- and far-future timeframes, respectively. These frequent severe events in the future are predominantly concentrated in EAS grids. While severe persistent drought events (return periods greater than 50 years) remain rare in SAS (except for central India) for SSI < -0.8, events with SSI < -0.5 could still occur frequently (return periods of less than 20 years) across the region. Consistent with the reduced propagation probability (as defined by CP) observed in mainland SEA in the far-future, persistent droughts in this region are rare, regardless of the SSI threshold.

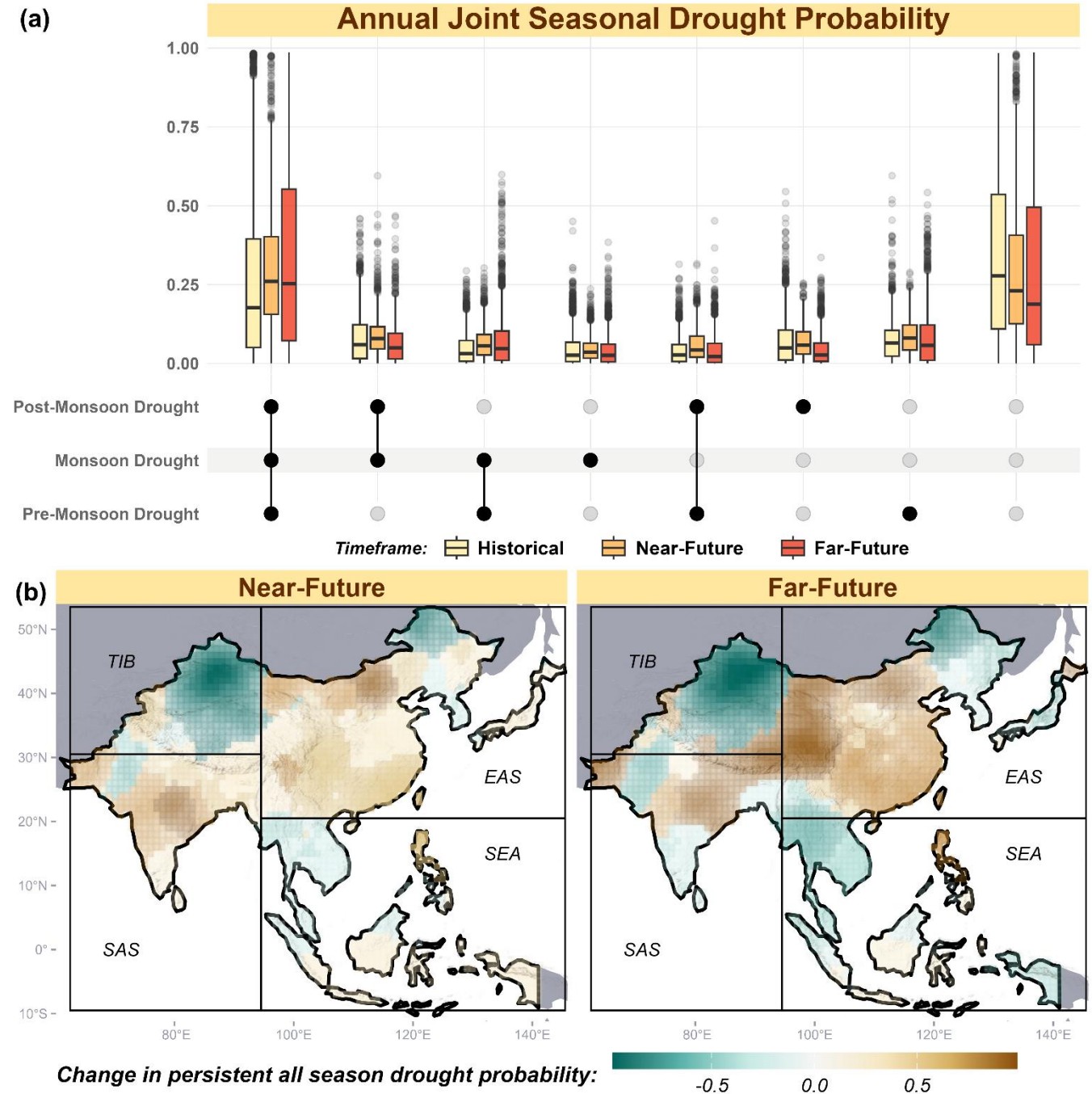

Figure 8: (a) Joint probabilities of all inter-seasonal drought scenarios across the study area for three timeframes, and (b) maps showing differences in all-season drought probabilities between future and historical timeframes.

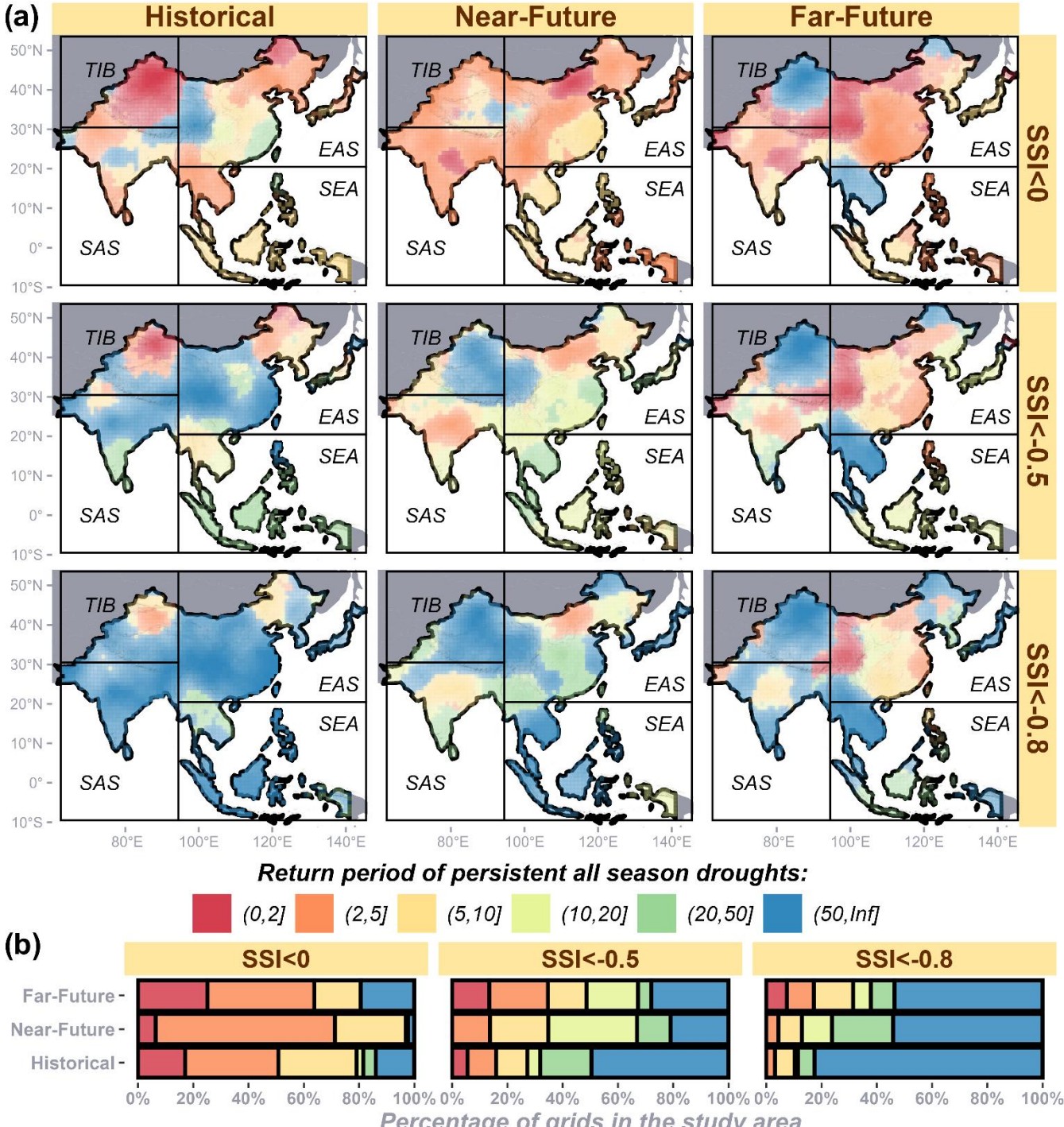

Figure 9: Return periods of persistent all-season droughts for various agricultural drought severity levels across timeframes, shown as (a) maps and (b) percentages of the study area.

**4.3 Drought spatial concurrence**

Considering the spatial disparities in the propagation and persistence of future droughts, the synchrony (or lack thereof) of monsoonal droughts across regions is examined to understand interconnected risks. The annual concurrence of drought between region pairs is assessed by examining the agricultural drought during monsoon months in each region. Figure 10 illustrates the probabilities of droughts exceeding a given spatial coverage threshold (k) in one region (Reg2) under the condition that the same spatial coverage threshold is exceeded in another region (Reg1). For instance, when 25% of EAS (Reg1) is under drought, there is a 62% probability that 25% of SAS (Reg2) will also experience droughts in the historical

timeframe. This probability increases to approximately 75% and 90% in the near- and far-future timeframes, respectively. Considering the dense populations and significant agricultural activity in these regions, the high spatial concurrence of future monsoonal droughts is concerning. In contrast, the conditional probability of drought concurrence between EAS (Reg1) and SEA (Reg2) decreases from 0.57 in the historical timeframe to 0.48 in the near-future, and further to 0.27 in the far-future. A similar reduction is observed between SAS (Reg1) and SEA (Reg2). For TIB (Reg2), the conditional probability of droughts

exceeding 25% spatial coverage in other regions (Reg1) remains similar between the historical and far-future timeframes but is significant lower in the near-future.

Overall, SAS and EAS are projected to experience an increase in simultaneous monsoonal droughts, while SEA shows reduced concurrence with the two regions. The spatial concurrent joint return periods are estimated and compared across timeframes using a historical reference event for these regions (Fig. 11). For example, a drought event affecting 36% of EAS

and 38% of SAS in a given year currently has a return period of approximately 30 years. However, such an event is expected to occur every 10 years in the near-future and less than every 5 years in the far-future, indicating more frequent spatially concurrent droughts between EAS and SAS. Large-scale widespread droughts affecting 50% of both EAS and SAS have return periods between 20 and 50 years in the future, which might not occur in the other timeframes. In contrast, the same historical reference event, which currently has a return period of 10 to 20 years for EAS and SEA, is projected to occur only once every

50 years or more in the future timeframes, highlighting the reduced likelihood of spatially concurrent droughts involving the SEA region. Similar patterns of reduced likelihood of spatially concurrent droughts in the far-future could happen between SAS and SEA. Specifically, widespread droughts become rare in SEA during far-future timeframes, such that droughts covering 40% of SEA could occur once in more than 50 years compared to more frequent events in other timeframes. Although most grids in TIB are not agriculturally productive, several rivers in the other three regions originate in this region, and hence,

analysing spatial drought concurrence involving TIB has hydrological significance (Fig. S11). The historical reference event of simultaneous droughts covering 31% and 42% of TIB and EAS has a return period of about 10 years in the historical timeframe, which becomes rare in the near-future (about 50 years return period). However, such events could once again become frequent with 2 to 5 years return period in the far-future. This pattern of spatially concurrent widespread drought events becoming rare in the near-future but frequent in the far-future is also noticed from the return period profile between

TIB and SAS regions. TIB-SEA has reduced future spatial concurrence similar to the return period profiles in the SEA-EAS and SEA-SAS region pairs.

   Although SPI-driven propagation values were not directly used to calculate the spatial concurrence of agricultural droughts (only SSI value were considered), several key findings align with those from the propagation analysis. For instance, concurrent cross-regional droughts between South Asia (SAS) and East Asia (EAS) are projected to increase in the far-future
timeframe (Fig. 11). In contrast, Southeast Asia (SEA) is expected to experience more non-synchronous (i.e., decreased concurrence) droughts with SAS and EAS in the future compared to the historical period. These findings are consistent with the trends in propagation probabilities: both SAS and EAS show increased propagation from meteorological to agricultural droughts in the future (Fig. 4(a)), while SEA shows a decline in propagation probability in the far-future relative to the historical timeframe. Thus, the results from the propagation and concurrence analyses are in agreement.

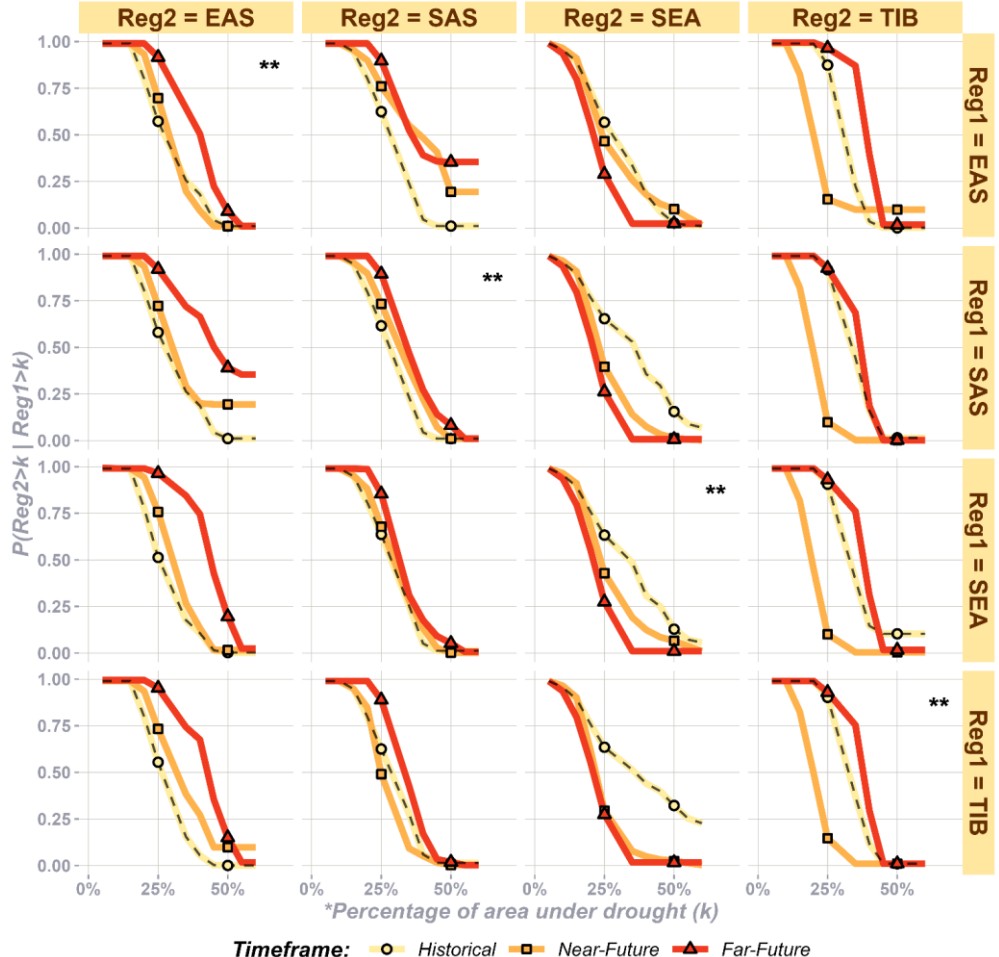

**Figure 10: Probability of a region exceeding different drought spatial coverage (k), conditioned on similar spatial coverage in another region, compared across timeframes.**

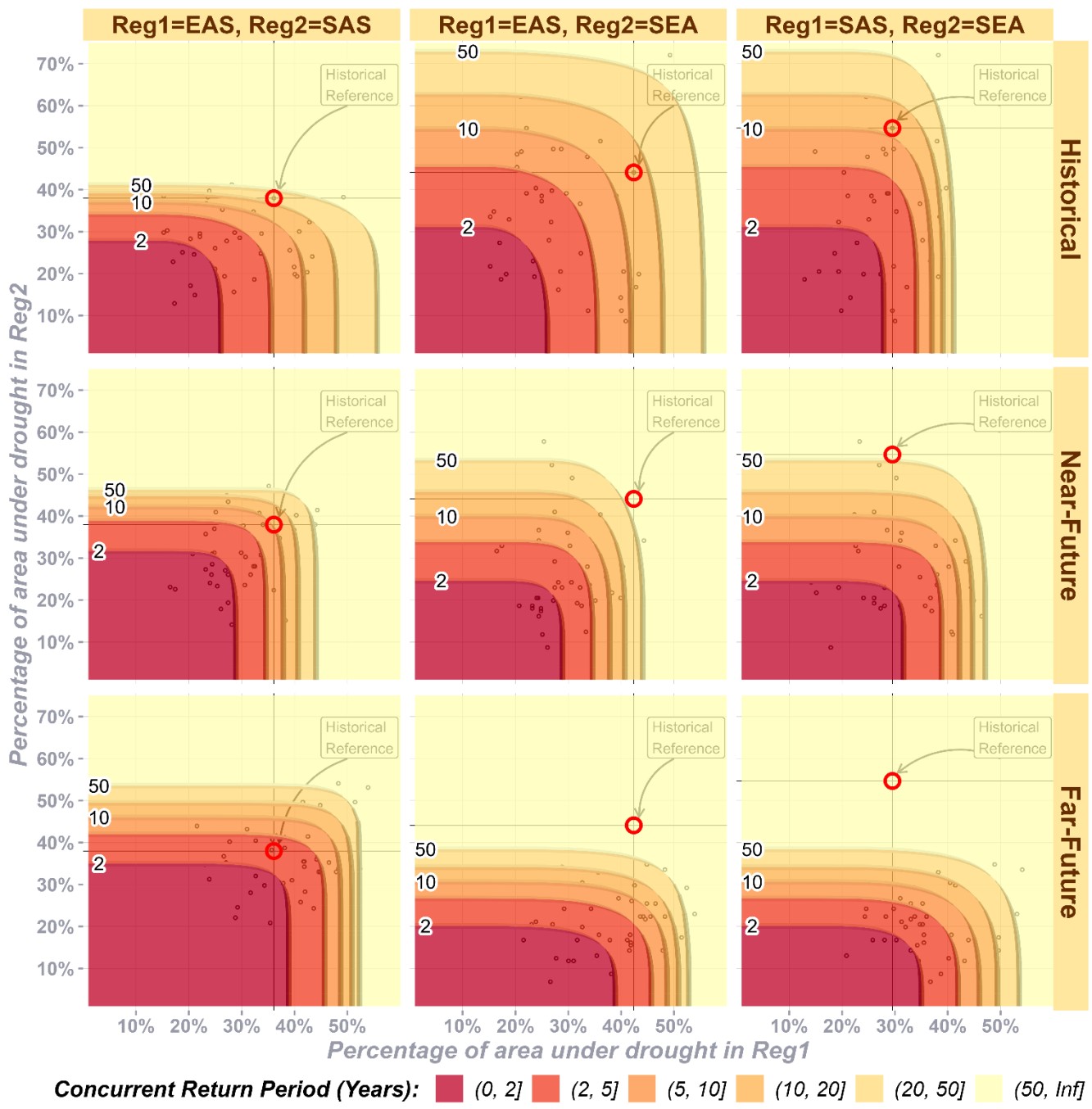

**Figure 11: Spatial concurrent return periods between region pairs across timeframes, with random variables (percentage of area under drought annually) shown as black dots.**

## 4.4 Further Discussion

While previous studies on drought propagations often focus on the transition of precursor meteorological droughts to agricultural or hydrological droughts, the persistence and spatial concurrence of the antecedent drought forms are often overlooked. The current study, however, highlights a strong correlation between the propagation process (CP) and the persistence of agricultural droughts across seasons (JP) in the future (Fig. 12a). Differences between future timeframes and historical CP values were compared with corresponding JP values for all grids using scatterplots. The strong correlation

(approximately 0.85, $p < 0.001$) indicates that grids experiencing accelerated propagation are also prone to persistent all-season droughts, and vice versa, in the future. In the far-future timeframe, a cluster of grids along the southeastern margin of the Tibetan Plateau (90°E to 104°E and 28°N to 35°N) exhibits significantly higher propagation and persistence compared to the historical timeframe (Fig. 12b). This cluster, often referred to as the "Asian Water Tower" (Leng et al., 2023), is of hydrological significance, as it serves as the origin for major rivers such as Brahmaputra (Yarlung Tsangpo), Irrawaddy, and Salween. It

also overlaps with the upper reaches of the Mekong, Yellow, and Yangtze rivers, which are critical for agriculture, industry, and livelihoods in both EAS and SAS regions. The strong negative correlation between soil moisture and temperature (Fig. S12) under future climate change scenarios contributes to the accelerated propagation and persistence of droughts in this region. These findings align with reports by Li et al. (2022) and Zhang et al. (2023), which document climate-change-induced terrestrial water deficits in Asia's Water Tower under the SSP5-8.5 scenario, further supporting the current results.

The projected increase in future spatial drought concurrence between EAS and SAS could have significant implications, including synchronized crop failures (Mehrabi and Ramankutty, 2019; Muthuvel and Sivakumar, 2024a), challenges in transboundary water management (Williams, 2018), and energy-sharing difficulties (Lv et al., 2024). These risks are further exacerbated by the projected rise in the exposed population within these regions, which host some of the world's megacities and major crop belts (Das et al., 2022; Zhao et al., 2022). In contrast, Mainland Southeast Asia is expected to

exhibit low propagation and persistence of droughts in future timeframes (Fig. 12b), supported by increases in precipitation extremes reported in previous climate change studies (Ge et al., 2021; Supharatid et al., 2022). This may contribute to its annual spatial asynchrony with EAS and SAS. While the increasing precipitation in SEA highlights the need for improved infrastructure to mitigate disasters and enhance water storage, it also presents opportunities for increased agricultural productivity. Such productivity could facilitate strategic virtual water trade (the hidden flow of water in traded commodities)

among neighboring countries by revising existing regional trade agreements (Chen et al., 2024), offering mutual benefits to the involved nations. Thus, the finding of drought synchrony between regions has some severe consequences, while

asynchrony          provides          an          opportunity          for          mutual          benefits.

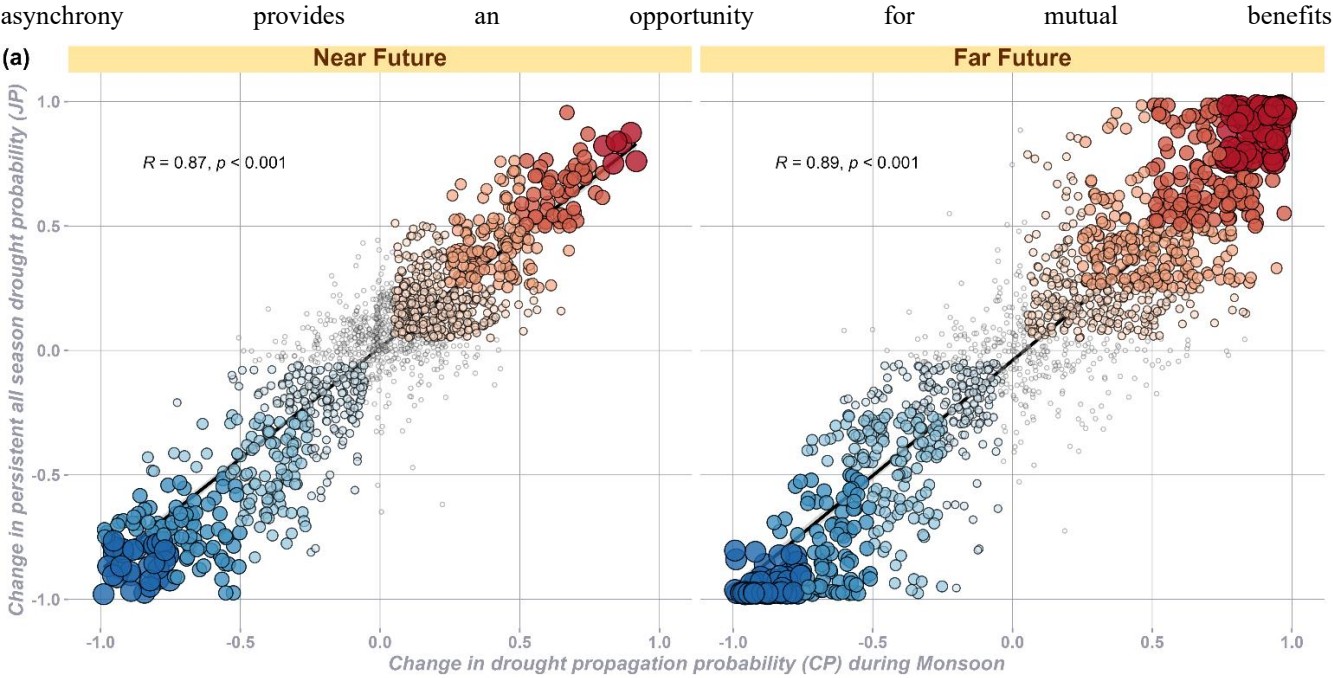

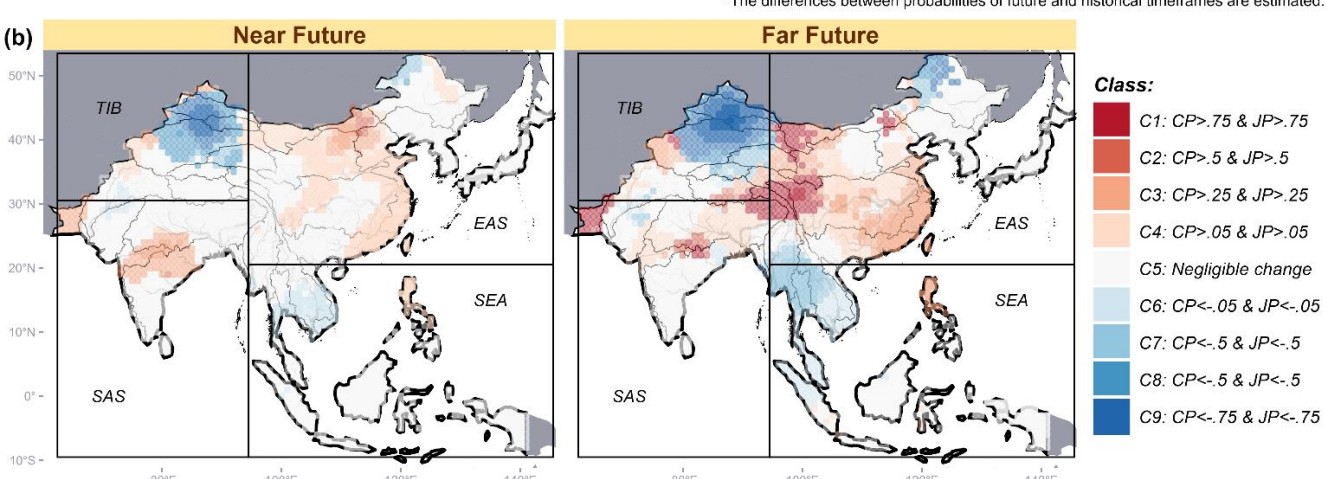

**Figure 12: (a)** Relationship between drought propagation and persistence using changes in their probability values between future and historical timeframes, and **(b)** spatial maps showing bivariate classification based on future persistence and propagation values to identify vulnerable grids.

While the present study provides a comprehensive assessment of future droughts by analysing key characteristics such as propagation, persistence, and spatial concurrence, there are a few limitations and opportunities for further improvement. The propagation duration in this study is determined using the correlation between meteorological and agricultural drought indices with a time lag. This linear approach may oversimplify the complex nature of drought propagation. Advanced techniques, such as the phase-space reconstruction method employed by Zhao et al. (2023), could be explored in

future studies, despite their higher computational demands. Additionally, the study focuses on temporal propagation but could be expanded to include spatial propagation assessments. Recent works (Mondal et al., 2023; Muthuvel and Sivakumar, 2024b) that apply network theory to identify spatial drought sources have shown promise in improving the key drivers of propagation.

Future research could incorporate large-scale teleconnections, as Long et al. (2024) demonstrated that these can significantly influence and accelerate the propagation process. Finally, as a data-driven study, the uncertainties associated with climate model projections may impact the results, even though a multi-model ensemble was used. Addressing these uncertainties through advanced bias correction techniques or alternative ensemble methods could further enhance the robustness of the findings. However, since this work focuses on drought propagation, it is essential to preserve the interrelationship between

precipitation and soil moisture. Traditional univariate bias correction techniques correcting individual variables (precipitation and soil moisture) could distort the time lags involved in propagation and weaken their correlation. That said, multivariate bias correction techniques, as explored in recent studies, could be helpful for analysing future drought propagation (Dieng et al., 2022). These methods preserve the inherent relationships between corrected variables, which is crucial for studying extreme events driven by multiple factors (Zscheischler et al., 2019). For example, Meng et al. (2022) applied multivariate bias

correction between precipitation and temperature to analyse compound dry and hot events. Similarly, applying such techniques to precipitation and soil moisture could enhance the study of drought propagation dynamics.

## 5 Conclusions

This study investigates the impact of climate change on drought propagation from meteorological to agricultural droughts, the persistence of inter-seasonal agricultural droughts, and their spatial concurrence across monsoon-dominant Asian

regions using a copula-based probabilistic approach. While future month-wise propagation durations are expected to follow the historical monsoon pattern, an accelerated propagation rate, indicated by conditional probability, is projected in South Asia (except Western and Peninsula India) and Eastern China, increasing vulnerability to frequent agricultural droughts. Random forest models identify temperature rise as the primary driver of agricultural droughts, with the Tibetan Plateau experiencing increased propagation and persistent all-season agricultural droughts in the far-future. In addition to the increase in

meteorological drought-driven agricultural droughts, there will also be an increase in non-rainfall-related agricultural droughts in the far-future, which could be attributed to increasing temperatures. Additionally, populous regions in South and East Asia may face frequent simultaneous monsoonal agricultural droughts, exacerbating water stress and crop losses, while Southeast Asia could experience reduced spatial concurrence of droughts, potentially fostering more collaborative virtual water transfer. This study enhances drought research by examining the interrelated aspects of temporal propagation, persistence, and spatial

concurrence, revealing a strong connection between drought propagation and persistence in future scenarios. While acknowledging limitations such as climate model uncertainty and linear assumptions in propagation duration calculations, the findings offer valuable insights for drought planning and adaptation in the context of climate change.

*Code and data availability:* Freely available open-access data have been used in the study and analysed using R coding language and are cited appropriately in the article.

*Author contributions:* DM and XQ conceptualized and developed the methodological framework. DM wrote the code and drafted the manuscript. XQ revised and edited the manuscript.

*Competing interests:* The authors declare that they have no conflict of interest.

*Financial support:* This research is supported by the Ministry of Education, Singapore, under its MOE Academic Research Fund Tier 3 (Award number MOE-MOET32022-0006). This work is also supported in part by the Ministry of Education, Singapore, under its Academic Research Fund Tier 1 (Grant No. RG147/24 and RG72/22). Any opinions, findings and conclusions or recommendations expressed in this material are those of the authors and do not reflect the views of the Ministry of Education, Singapore.

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
