# Peer review of "Probabilistic Analysis of Future Drought Propagation, Persistence, and Spatial Concurrence in Monsoon-Dominant Asian Region under Climate Change"

_EGUsphere, 2025_

## Author Comment (AC1)

**General Comments:**

The authors did a thorough analysis of the future droughts in Monsoon-dominant Asian under the worst-case emission scenario of SSP5-8.5. The analysis of propagation from meteorological to hydrological droughts is new, and the use of bivariate copula function for analyzing drought propagation and spatial concurrence is interesting. I only have a few comments as listed below.

**Authors' Response:** We thank the reviewer for taking time to review our work and for providing valuable comments to improve the manuscript. We have tried to respond to each of the comments and will incorporate the suggested changes in the revised version. Addressing these comments in the revised version will surely enhance the quality of the manuscript immensely.

**Specific comments:**

**Reviewer Comment #1:** Abstract: since the propagation from meteorological drought to agricultural drought is a highlight of this work. It's helpful to indicate in the abstract that meteorological and agriculture droughts are measured using SPI and SSI, respectively.

**Authors' Response #1:** We agree that indicating the indices of SPI and SSI explicitly in the abstract is necessary and will do so in the revision version.

**Reviewer Comment #2:** Line 61: I believe there are quite a few studies on drought analysis under climate change. I am not sure whether "only" is the most accurate or appropriate term in this case. Citing these works can help readers better understand the current state of research on this topic.

**Authors' Response #2:** We regret this oversight. There are indeed some seminal works on the propagation of meteorological to agricultural droughts. We will revise the sentence to clarify that existing drought propagation studies generally do not examine the repercussions of propagation, specifically the persistence and concurrence of agricultural droughts.

Additionally, we will cite the recent seminal works on meteorological to agricultural drought propagation (Dai et al., 2022; Ding et al., 2021; Fawen et al., 2023; Xu et al., 2023). Accordingly, we propose to include the following revised sentence:

*"While several studies have examined the propagation from meteorological to agricultural droughts (Dai et al., 2022; Ding et al., 2021; Fawen et al., 2023; Xu et al., 2023), their repercussions, such as the persistence and cross-regional concurrence of agricultural droughts, are often overlooked."*

**Reviewer Comment #3:** Lines 155-160, a common practice in climate impact studies is to use bias correction techniques and correct the biases in GCM output before any further analysis. Do you think this can help reduce the errors in Figure 2c and Fig. S1? How about the difference between observation and GCM output in Fig. S2?

**Authors' Response #3:** We thank the reviewer for bringing this up. We agree that bias correction can improve the data quality by minimizing errors. However, since this work focuses on drought propagation, it is essential to preserve the interrelationship between precipitation and soil moisture. We refrained from using traditional univariate bias correction techniques, as correcting individual variables (precipitation and soil moisture) could distort the time lags involved in propagation and weaken their correlation.

That said, multivariate bias correction techniques, as explored in recent studies, could be helpful for analyzing future drought propagation (Dieng et al., 2022). These methods preserve the inherent relationships between corrected variables, which is crucial for studying extreme events driven by multiple factors (Zscheischler et al., 2019). For example, Meng et al. (2022) applied multivariate bias correction between precipitation and temperature to analyze compound dry and hot events. Similarly, applying such techniques to precipitation and soil moisture could enhance the study of drought propagation dynamics.

In the revised manuscript, we will explicitly acknowledge the limitation of not employing a multivariate bias correction approach in the current study. Nonetheless, the Multi-Model Ensemble (MME) used here still performs well. In response to the reviewer's suggestion, we will include monthly plots showing the mean difference between GCM data (Multi Model

Mean using Bayesian Model Averaging) and GLDAS (Observed data). We will also incorporate maps comparing drought properties from historical GCM data and GLDAS.

**Reviewer Comment #4:** Line 245: why is R^2 the only performance metrics for soil moisture prediction? With some R^2 values lower than 0.5 in the results, how to justify the accuracy of the RF model or the reliability of its feature importance results?

**Authors' Response #4:** We will add a map showing RMSE values to complement the existing $R^2$-themed maps in validating the random forest (RF) models. These RF models predict time series of soil moisture using temporally varying climatic predictors to understand their influence. A total of 5,364 (1,788 grids x 3 timeframes) RF models were developed, one for each grid across three timeframes. Apart from the climatic variables, other grid-specific predictors such as elevation (Zhang et al., 2024) and climate characteristics (Zhang et al., 2021), which play a crucial role, are not included in these RF models. Since these variables are temporally constant for a given grid, they do not affect the time series prediction. Consequently, at certain grid points with low $R^2$ values, factors beyond the selected climate variables may influence soil moisture.

Despite low $R^2$ values in some cases, the variable with the highest feature importance still demonstrates a relatively stronger influence to the other climate predictors. To complement these temporal RF models, which help identify key climatic drivers of soil moisture, spatial RF models will be developed. These will incorporate grid-specific predictors, such as elevation and climate classification, along with aggregated climate variables to predict propagation probability. Please refer to **Author's Response #5** for more details on these spatial RF models.

**Reviewer Comment #5:** Line 250: The predictors that are important in the RF model, are the ones that are important for the estimation of soil moisture. Are they necessarily the same as the ones that may lead to soil moisture deficit? How could the feature importance results be best interpreted?

**Authors' Response #5:** We thank the reviewer for the insightful comment. We understand the concern that the important variables identified in the temporal RF models predicting soil

moisture do not explicitly relate to soil moisture deficit. To address this, and to complement the grid-specific temporal RF models, we plan to include spatial RF models that directly predict drought propagation probabilities (P(SSI<-0.5|SPI<-0.5)) for each timeframe (three RF models in total). These spatial models aims to address the reviewer's concern by using soil moisture deficit, expressed as a conditional probability, as the predictand. Hu et al. (2024) developed similar spatial RF models to assess the relative importance of variables in drought propagation. In our case, these are considered spatial models because each grid has a single propagation probability value, which forms one row in the training dataset.

The predictors for these models will include elevation, climate zones, and the monthly means of climate variables (i.e. temperature, precipitation, humidity, vegetation cover, solar radiation at 100 m, and wind speed). Since the propagation probability is computed over an entire timeframe, the climate variables are aggregated as monthly means for each timeframe. We intend to retain the results from the temporal RF models, as they complement the spatial models. The spatial RF models capture the influence of stationary predictors (e.g. elevation and climate zone), which the temporal models cannot. Conversely, the temporal RF models (5,364 in total: 1,788 grids x 3 timeframes) capture temporal climatic variations, where each month's data forms a row of training data for a given grid-specific model. These variations must be aggregated in the spatial RF models.

**Technical corrections:**

**Reviewer Comment #6:** Line 100: the "+" sign suggests a summation of these variables inside the function, which is not a rigorous expression. Since SM depends on these variables separately, the notation f(T, Pr, H, VC, SR100, W, TS) would be more appropriate.

**Authors' Response #6:** We thank the reviewer for this comment and agree that it is more appropriate for the RF model to be denoted as SM = f(T, Pr, H, VC, SR, W, TS), instead of using "+" sign. We will incorporate this in the revised version.

**References**

Dai, M., Huang, S., Huang, Q., Zheng, X., Su, X., Leng, G., Li, Z., Guo, Y., Fang, W., and Liu, Y.: Propagation characteristics and mechanism from meteorological to agricultural drought in various seasons, J Hydrol (Amst), 610, https://doi.org/10.1016/j.jhydrol.2022.127897, 2022.

Dieng, D., Cannon, A. J., Laux, P., Hald, C., Adeyeri, O., Rahimi, J., Srivastava, A. K., Mbaye, M. L., and Kunstmann, H.: Multivariate Bias-Correction of High-Resolution Regional Climate Change Simulations for West Africa: Performance and Climate Change Implications, Journal of Geophysical Research: Atmospheres, 127, https://doi.org/10.1029/2021JD034836, 2022.

Ding, Y., Gong, X., Xing, Z., Cai, H., Zhou, Z., Zhang, D., Sun, P., and Shi, H.: Attribution of meteorological, hydrological and agricultural drought propagation in different climatic regions of China, Agric Water Manag, 255, https://doi.org/10.1016/j.agwat.2021.106996, 2021.

Fawen, L., Manjing, Z., Yong, Z., and Rengui, J.: Influence of irrigation and groundwater on the propagation of meteorological drought to agricultural drought, Agric Water Manag, 277, https://doi.org/10.1016/j.agwat.2022.108099, 2023.

Hu, C., Xia, J., She, D., Wang, G., Zhang, L., Jing, Z., Hong, S., and Song, Z.: Precipitation exacerbates spatial heterogeneity in the propagation time of meteorological drought to soil drought with increasing soil depth, Environmental Research Letters, 19, https://doi.org/10.1088/1748-9326/ad4975, 2024.

Meng, Y., Hao, Z., Feng, S., Guo, Q., and Zhang, Y.: Multivariate bias corrections of CMIP6 model simulations of compound dry and hot events across China, Environmental Research Letters, 17, https://doi.org/10.1088/1748-9326/ac8e86, 2022.

Xu, Z., Wu, Z., Shao, Q., He, H., and Guo, X.: From meteorological to agricultural drought: Propagation time and probabilistic linkages, J Hydrol Reg Stud, 46, https://doi.org/10.1016/j.ejrh.2023.101329, 2023.

Zhang, C., Han, Z., Wang, S., Wang, J., Cui, C., and Liu, J.: Accelerated Atmospheric to Hydrological Spread of Drought in the Yangtze River Basin under Climate, Remote Sens (Basel), 16, https://doi.org/10.3390/rs16163033, 2024.

Zhang, H., Ding, J., Wang, Y., Zhou, D., and Zhu, Q.: Investigation about the correlation and propagation among meteorological, agricultural and groundwater droughts over humid and arid/semi-arid basins in China, J Hydrol (Amst), 603, https://doi.org/10.1016/j.jhydrol.2021.127007, 2021.

Zscheischler, J., Fischer, E. M., and Lange, S.: The effect of univariate bias adjustment on multivariate hazard estimates, Earth System Dynamics, 10, 31–43, https://doi.org/10.5194/esd-10-31-2019, 2019.

---

## Author Comment (AC2)

General Comments:

The authors investigate the impacts of climate change on drought propagation from meteorological to agricultural droughts in monsoon-dominant Asian regions. Understanding drought propagation mechanisms under climate change is crucial, particularly in assessing temporal transitions in drought propagation, persistence, and spatial concurrence.

**Authors' Response:** First of all, we thank the reviewer for taking time to review our article. The reviewer has raised several valid and pertinent questions, which we have attempted to address. Incorporating these modifications and clarifications in the revised version will greatly improve the manuscript.

While the study addresses an important research gap, several critical issues undermine the validity of the findings:

**Reviewer Comment #1:** The manuscript does not adequately justify the need to study drought propagation, persistence, and spatial concurrence together. Additionally, the claim that studies on meteorological-to-agricultural drought propagation are lacking is inaccurate, as several previous studies already exist (Ding et al., 2021; Dai et al., 2022; Fawen et al., 2023; Xu et al., 2023). Lastly, since this study also focuses on a regional scale, the authors should explicitly clarify its novel contributions and regional significance.

**Authors' Response #1:** We appreciate the reviewer for bringing this up and would like to clarify the overall structure and contribution of the work. The first part of the study focuses on understanding the propagation from meteorological to agricultural droughts using temporal lags and propagation strengths. This analysis helps in forecasting impending agricultural droughts, thereby aiding preparedness efforts. After establishing the relationship between precursor meteorological and antecedent agricultural droughts, the subsequent sections on drought persistence and cross-regional concurrence shift the focus to agricultural droughts. This is because soil moisture deficits (i.e., agricultural droughts) directly impact crop and vegetation growth (Modanesi et al., 2020).

Regarding the novel contributions of this study, while most existing works focus on the propagation process, they often do not examine the subsequent persistence and spatial concurrence of agricultural droughts. In contrast, the present study aims to analyse the

persistence and spatial concurrence of agricultural droughts (using SSI directly) and explore their potential interrelationship with the initial propagation process. Our work attempts to access these crucial aspects holistically, rather than examining them in isolation.

We apologize for the earlier oversight in stating that studies on meteorological-to-agricultural drought propagation are lacking. We acknowledge that several seminal works do exist on this topic. We will rephrase the relevant sentence to indicate that existing studies primarily focus on the propagation process and tend to overlook its repercussions, namely the persistence and spatial concurrence of agricultural droughts. We will also cite the relevant seminal works on meteorological-to-agricultural drought propagation as mentioned by the reviewer (Dai et al., 2022; Ding et al., 2021; Fawen et al., 2023; Xu et al., 2023).

**Reviewer Comment #2:** Unclear Monsoon-Based Classification for Drought Persistence: The rationale for categorizing drought persistence into pre-monsoon, monsoon, and post-monsoon periods is not well-explained. It remains unclear whether this classification is tied to SPI-SSI propagation dynamics or solely based on SSI thresholds (e.g., SSI < -0.5). A stronger theoretical or empirical basis for this approach is needed. Justify the focus on monsoonal seasons for drought propagation implications.

**Authors' Response #2:**

We deeply appreciate the reviewer's concern on the monsoon-based classification, and offer the following justification:

**Rationale for categorizing seasons to analyse drought persistence:**

Crops are cultivated across multiple seasons throughout the study area, with different stages of crop growth aligning with the pre-monsoon, monsoon, and post-monsoon seasons. Adequate soil moisture during all these periods is critical for crop development. For example, in China, spring crops are typically sown in May and harvested around October, while winter crops like wheat are planted in September and harvested by following June (Li and Lei, 2021). Similarly, in South Asia, different crops are grown in three different phases including Zaid, Kharif, and Rabi, that correspond to the pre-monsoon, monsoon, and post-monsoon seasons, respectively (Joseph and Ghosh, 2023). The pre-monsoon season is particularly important for rain-fed

agriculture, especially during early crop planting in May. Soil moisture at the end of the monsoon season (September) aligns with the heading stage of Kharif crops such as rice. Both the pre-monsoon (May) and post-monsoon (December) months also coincide with the initial stages of planting for summer (monsoonal) and winter (Rabi) crops like wheat. While monsoonal droughts are a key focus, given that these months account for the majority of annual rainfall, droughts during the pre- and post-monsoon periods can also severely impact crop growth.

Yang et al. (2021) highlight that soil moisture deficits prior to planting can impair seedling root development, significantly affecting crop yields. Thus, droughts in May (pre-monsoon) that coincide with land preparation and early sowing stages must also be analysed. Moreover, residual soil moisture from the monsoon is crucial for winter crop growth during the post-monsoon season (represented by December soil moisture). Ford and Labosier (2014) define drought persistence as the tendency of drought to continue across seasons. When drought conditions persist across the end of pre-monsoon, monsoon, and post-monsoon periods, the risk of widespread crop failures increases. Prior studies (Fang et al., 2019; Swain et al., 2024) have used bivariate copulas to examine the intra-seasonal drought between dry and wet seasons. In this study, we extend this approach using a trivariate copula (with SSI values from the end months of the pre-monsoon, monsoon, and post-monsoon seasons) to better understand drought persistence across the key agricultural seasons in the region.

Drought persistence in this study is calculated directly using a soil moisture-based index (SSI at a monthly timescale), which reflects agricultural droughts. It is not derived from SPI values (meteorological droughts) or drought propagation probability (defined by Conditional Probability, CP). However, as shown in Fig. 10, there is a strong linear relationship between drought persistence (defined by Joint Probability, JP) and drought propagation (CP). In Fig. 10, changes in CP values between future and historical timeframes were compared with the corresponding JP values across all grids using scatterplots and thematic maps. This highlights regions with high drought persistence. The strong correlation (approximately 0.85, $p < 0.001$) suggests that grids experiencing accelerated drought propagation are also likely to face persistent droughts across seasons in the future, and vice versa. Further details explaining the rationale for explicitly studying drought persistence and spatial concurrence using SSI values are provided in our next response (**Authors' Response #3**). We kindly refer the reviewer to that section.

**Reviewer Comment #3:** Ambiguity in Drought Concurrence Analysis: The assessment of spatial drought concurrence relies on SSI thresholds but does not explicitly link to SPI-driven propagation. The authors should clarify whether the observed concurrence reflects independent agricultural droughts or is influenced by meteorological drought propagation.

**Authors' Response #3:** We thank the reviewer for the observation and will clarify the ambiguity regarding the drought concurrence analysis. As mentioned in earlier responses, the persistence and spatial concurrence of agricultural droughts are not explicitly based on SPI-driven propagation measures in their methodological computation. However, the resulting patterns from both persistence and spatial concurrence analyses are compared with propagation probabilities to understand their influence.

Although SPI-driven propagation values were not directly used in calculating the spatial concurrence of agricultural droughts (only SSI value were considered), several key findings align with those from the propagation analysis. For instance, our results indicate an increase in concurrent cross-regional droughts between South Asia (SAS) and East Asia (EAS) in the far-future timeframe (Fig. 9). In contrast, Southeast Asia (SEA) is projected to experience more non-synchronous (i.e., decreased concurrence) droughts with SAS and EAS in the future compared to the historical period. These findings are consistent with the trends in propagation probabilities: both SAS and EAS show an increase in propagation from meteorological to agricultural droughts in the future (Fig. 4(a)), while SEA shows a decrease in propagation probability in the far-future compared to the historical timeframe. Thus, the results from the propagation and concurrence analyses are in agreement. These consistencies suggest that propagation characteristics (from meteorological to agricultural droughts) play a significant role in shaping the persistence (Fig. 10) and spatial concurrence of subsequent agricultural droughts.

**Rationale for using agricultural droughts (SSI) directly to analyse drought persistence and spatial concurrence:**

The reviewer's 2$^{nd}$ and 3$^{rd}$ comments raise a common concern about whether the agricultural drought indices used in analysing drought persistence and concurrence are directly influenced by meteorological drought propagation. To address this, we computed the probability of meteorological droughts occurring under the condition of existing agricultural droughts (P(SPI<0|SSI<-0.5)) across three timeframes, and compared it with the forward propagation

probability (P(SSI<0|SPI<-0.5)) shown in **Fig. R1**. This reverse propagation probability, (P(SPI<0|SSI<-0.5)), reflects the extent to which an agricultural drought is linked to a prior meteorological drought. The density plots show a decline in reverse propagation probability values in both the near- and far-future scenarios relative to the historical period across all regions, with the most significant decrease observed in EAS in the far-future timeframe. This suggests that the number of agricultural drought events not directly attributable to meteorological droughts is expected to rise.

Simultaneously, the forward propagation probability (P(SSI<0|SPI<-0.5)) indicates that meteorological droughts are increasingly driving agricultural droughts in SAS and EAS in the future (Fig. 4). Thus, while meteorological-driven agricultural droughts are projected to increase, so too are those unrelated to rainfall deficits. Random forest models used to predict soil moisture further reveal that temperature becomes the dominant driver of agricultural droughts across more than 50% of the study area in future scenarios, compared to about 25% in the historical period (Fig. 5(c)). This supports the observed increase in non-rainfall-related agricultural droughts. Under the SSP5-8.5 scenario, significant temperature increases are expected towards the end of the century (Qiao et al. 2023). This warming could intensity soil moisture deficits, leading to a shift from meteorological-driven to temperature-driven agricultural droughts. Additionally, above-average rainfall (SPI>0) may occur in the form of short-term, intense storms, which may not adequately replenish soil moisture under higher temperatures, which is a situation exacerbated by climate change. These findings support the methodological decision to use SSI directly to assess drought persistence and concurrence, rather than relying solely on SPI. In EAS, and SAS, although meteorological-driven agricultural droughts are projected to rise in the far-future, they are likely to be more severe due to the increasing influence of temperature, potentially resulting in more frequent compound drought-heatwave events.

We hope these new insights address the methodological concerns raised and justify the direct use of SSI in our analysis. We intend to incorporate these findings in the revised manuscript.

[Figure]

**Figure R1.** Reverse propagation probability (P(SPI<0|SSI<-.5)) indicates if an agricultural drought is linked to a prior meteorological drought.

**Reviewer Comment #4:** L93: Random Forest Model Design: The use of soil moisture as the predictand (rather than drought propagation metrics) limits the model's ability to identify key drivers of propagation. Restructuring the model to treat propagation as the predictand (with climatic variables as predictors) would better address the study's primary objective.

**Authors' Response #4:** We thank the reviewer for the insightful comment. We understand the concern that the important variables identified in the temporal RF models predicting soil moisture do not explicitly relate to drought propagation. To address this, we plan to complement the grid-specific temporal RF models with spatial RF models aimed at directly

predicting drought propagation probabilities (P(SSI<-0.5|SPI<-0.5)) for each timeframe (i.e., three separate RF models). These spatial RF models are designed to predict propagation probabilities at each grid, with each grid contributing one training data row per model. The predictors will include elevation, climate zones, and the monthly means of climate variables (i.e., temperature, precipitation, humidity, vegetation cover, solar radiation at 100 m, wind speed, and surface temperature). Since the propagation probabilities are computed over the entire timeframe, the climate variables are aggregated as monthly mean for the respective timeframe in the spatial models.

We intend to retain the temporal RF models as they offer complementary insights. While the spatial RF models account for the influcence of stationary predictors (such as elevation and climate zone), which temporal models cannot, the grid-specific RF models (1,788 grids x 3 timeframes = 5,364 models) can capable of capturing temporal variability, with each month's data contributing a separate row of training data. Moreover, results from the far-future temporal RF models further support the observed shift towards temperature-driven soil moisture droughts, as discussed in **Authors' Response #3**.

**Minor comments:**

**Reviewer Comment #5:** L13-L16: " In terms of the return period, all-season droughts that historically occurred once in more than 50 years could happen as frequently as every five years by the far-future (2061-2100) at the hydrologically significant Tibetan Plateau." The statement is confusing. Rephrase for clarity.

**Authors' Response #5:** We thank the reviewer for the comment and would like to rephrase the statement as follows:

*"At the hydrologically significant Tibetan Plateau, all-season droughts that were historically rare (with return periods exceeding 50 years) could occur as frequently as once every 5 years in the far-future period (2061-2100)."*

**Reviewer Comment #6:** L17-L19: "The spatial concurrence of monsoonal agricultural droughts between region pairs such as South Asia (SAS), East Asia (EAS), Southeast Asia

(SEA), and Tibetan region (TIB) was also assessed." Replace methodological descriptions (e.g., region pairs assessed) with concrete findings.

**Authors' Response #6:** We thank the reviewer for this comment and will replace this statement based on methodology with more appropriate findings in the abstract.

**Reviewer Comment #7:** L27: Cite specific literature comparing disaster impacts.

**Authors' Response #7:** We will add appropriate literatures for the specified statement in the revised version.

**Reviewer Comment #8:** L40-41: "While comparing propagation between basins of different climate zones, Zhang et al. (2021) found arid basin to have lower propagation durations compared to humid and sub-humid basins" Rephrase for clarity.

**Authors' Response #8:** We thank the reviewer for the suggestion and will rephrase the sentence for improved clarity as follows:

*"Climate characteristics are known to influence drought propagation. In this context, Zhang et al. (2021) found that arid basins tend to have shorter propagation durations compared to humid and sub-humid basins."*

**Reviewer Comment #9:** L64-65: "To address the aforementioned gaps, this study proposes a comprehensive copula-based multivariate probabilistic approach, utilizing climate model projection data." Separate the copula method and climate model applications to avoid logical gaps.

**Authors' Response #9:** We thank the reviewer for the suggestion and will rephrase the sentence for improved clarity as follows:

*"To address the aforementioned gaps, this study proposes a comprehensive copula-based multivariate probabilistic framework, which will be applied to climate model projection data."*

**Reviewer Comment #10:** 2.1.1: Describe monthly precipitation, SPI/SSI calculations, and soil moisture datasets.

**Authors' Response #10:** We thank the reviewer for the comment, and we will include a detailed description of monthly precipitation, SPI/SSI calculations, and soil moisture datasets in section 2.1.1.

**Reviewer Comment #11:** 2.1.3: Please provide a detailed description of the hyperparameters in the random forest model (e.g., the number of trees, maximum depth), explaining the parameter tuning process to validate the robustness of the model. Furthermore, it is recommended to include other evaluation metrics (e.g., RMSE) to more accurately demonstrate model performance. Please provide the cross-validation of RF models and shows the R2 of different regions.

**Authors' Response #11:** We appreciate the reviewer's suggestion regarding hyperparameter tuning and performance evaluation of the RF models. Due to the computational intensity involved in training 5364 (1788 grids X 3 timescales), we adopted a fixed set of fixed hyperparameters similar to the ones used by Dai et al. (2022). The hyperparameters are listed as follows:

- ntree = 500 (number of trees)
- mtry = 5 (number of variables tried at each split)
- nodesize = 5 (minimum size of terminal nodes)

In addition to the $R^2$ value (Fig. 5(a)), RMSE values will be plotted spatially for evaluation.

However, for the spatial RF models to predict drought propagation probabilities with timeframes (3 RF models), extensive grid-based hyperparameter tuning with cross-validation

will be performed in the revised version. The three RF models will be evaluated using metrics such as $R^2$ and RMSE.

**Reviewer Comment #12:** Fig.2b: Provide citations for the basis of regional divisions, which is crucial to spatial concurrence analysis.

**Authors' Response #12:** The regional divisions are based on thematic maps from works such as Giorgi and Bi (2009) and Sillmann et al. (2013). The citations will be added appropriately in the revised version.

**Reviewer Comment #13:** Fig. 2c: Compare GLDAS and GCM-derived SPI monthly drought characters (not average precipitation) to validate GCM reliability for drought propagation.

**Authors' Response #13:** We thank the reviewer for the insightful comment. We will prepare thematic maps of cumulative drought severities from GLDAS and GCM data.

**Reviewer Comment #14:** Datasets: Replace GLDAS precipitation/soil moisture with more reliable datasets (e.g., MSWEP for precipitation, GLEAM for soil moisture).

**Authors' Response #14:** MSWEP and GLEAM (with MSWEP as the major input) have been commonly used in drought propagation studies to minimize uncertainty arising from disparate data sources (Gupta and Karthikeyan, 2024; Odongo et al., 2023). For a similar reason, ensuring consistency between precipitation and soil moisture data, GLDAS was used in this study. GLDAS datasets are widely employed in drought research and have been shown to effectively capture major historical drought events. For instance, we plotted the percentage of areas affected by meteorological and agricultural droughts using GLDAS data (**Fig. R2**), which clearly reflects significant events such as the 2009 drought (Barriopedro et al., 2012).

Additionally, Gupta and Karthikeyan (2024) reported good agreement in meteorological drought characteristics across MSWEP, CHIRPS, and GLDAS, supporting the reliability of GLDAS. While MSWEP and GLEAM are excellent datasets for drought propagation studies,

existing literature suggests that the GLDAS performs comparably well. Furthermore, the three timeframes used in the study, historical (1975-2014), near-future (2021-2060, and far-future (2061-2100), were selected to ensure equal-length epochs of 40 years for consistent comparison of drought propagation, persistence, and spatial concurrence. However, MSWEP and GLEAM begin only in 1979 and 1980, respectively, making them unsuitable for this time range. Given these these considerations, we believe GLDAS is an appropriate choice for the present study.

[Figure]

**Figure R2.** The percentage of drought-affected areas (based on the percentage of grids falling at different ranges of SPI and SSI values) annually. Spatial maps of 2009 drought events.

**References**

Barriopedro, D., Gouveia, C. M., Trigo, R. M., and Wang, L.: The 2009/10 drought in China: Possible causes and impacts on vegetation, J Hydrometeorol, 13, 1251–1267, https://doi.org/10.1175/JHM-D-11-074.1, 2012.

Dai, M., Huang, S., Huang, Q., Zheng, X., Su, X., Leng, G., Li, Z., Guo, Y., Fang, W., and Liu, Y.: Propagation characteristics and mechanism from meteorological to agricultural drought in various seasons, J Hydrol (Amst), 610, https://doi.org/10.1016/j.jhydrol.2022.127897, 2022.

Fang, W., Huang, S., Huang, G., Huang, Q., Wang, H., Wang, L., Zhang, Y., Li, P., and Ma, L.: Copulas-based risk analysis for inter-seasonal combinations of wet and dry conditions under a changing climate, International Journal of Climatology, 39, 2005–2021, https://doi.org/10.1002/joc.5929, 2019.

Ford, T. and Labosier, C. F.: Spatial patterns of drought persistence in the Southeastern United States, International Journal of Climatology, 34, 2229–2240, https://doi.org/10.1002/joc.3833, 2014.

Giorgi, F. and Bi, X.: Time of emergence (TOE) of GHG-forced precipitation change hot-spots, Geophys Res Lett, 36, https://doi.org/10.1029/2009GL037593, 2009.

Gupta, A. and Karthikeyan, L.: Role of Initial Conditions and Meteorological Drought in Soil Moisture Drought Propagation: An Event-Based Causal Analysis Over South Asia, Earths Future, 12, https://doi.org/10.1029/2024EF004674, 2024.

Joseph, J. and Ghosh, S.: Representing Indian Agricultural Practices and Paddy Cultivation in the Variable Infiltration Capacity Model, Water Resour Res, 59, https://doi.org/10.1029/2022wr033612, 2023.

Li, J. and Lei, H.: Tracking the spatio-temporal change of planting area of winter wheat-summer maize cropping system in the North China Plain during 2001–2018, Comput Electron Agric, 187, https://doi.org/10.1016/j.compag.2021.106222, 2021.

Modanesi, S., Massari, C., Camici, S., Brocca, L., and Amarnath, G.: Do Satellite Surface Soil Moisture Observations Better Retain Information About Crop-Yield Variability in Drought Conditions?, Water Resour Res, 56, https://doi.org/10.1029/2019WR025855, 2020.

Odongo, R. A., De Moel, H., and Van Loon, A. F.: Propagation from meteorological to hydrological drought in the Horn of Africa using both standardized and threshold-based

indices, Natural Hazards and Earth System Sciences, 23, 2365–2386, https://doi.org/10.5194/nhess-23-2365-2023, 2023.

Qiao, L., Zuo, Z., Zhang, R., Piao, S., Xiao, D., and Zhang, K.: Soil moisture–atmosphere coupling accelerates global warming, Nat Commun, 14, https://doi.org/10.1038/s41467-023-40641-y, 2023.

Sillmann, J., Kharin, V. V., Zhang, X., Zwiers, F. W., and Bronaugh, D.: Climate extremes indices in the CMIP5 multimodel ensemble: Part 1. Model evaluation in the present climate, Journal of Geophysical Research Atmospheres, 118, 1716–1733, https://doi.org/10.1002/jgrd.50203, 2013.

Swain, S. S., Mishra, A., and Chatterjee, C.: Assessment of Basin-Scale Concurrent Dry and Wet Extreme Dynamics Under Multimodel CORDEX Climate Scenarios, International Journal of Climatology, https://doi.org/10.1002/joc.8677, 2024.

Yang, M., Wang, G., Lazin, R., Shen, X., and Anagnostou, E.: Impact of planting time soil moisture on cereal crop yield in the Upper Blue Nile Basin: A novel insight towards agricultural water management, Agric Water Manag, 243, https://doi.org/10.1016/j.agwat.2020.106430, 2021.

---

## Author Response (AR1)

Ms Ref. No.: egusphere-2025-522

Title: Probabilistic Analysis of Future Drought Propagation, Persistence, and Spatial Concurrence in Monsoon-Dominant Asian Region under Climate Change

Authors: Dineshkumar Muthuvel, Xiaosheng Qin

Dear Editor and Reviewers,

We would like to sincerely thank the Editor and both reviewers for their thoughtful and constructive comments on our manuscript. We are encouraged by their positive overall evaluation and have carefully revised the manuscript in response to all suggestions and recommendations. We believe that the revisions have significantly improved the clarity, quality, and presentation of our work. Please find the following documents attached for your consideration:

(1) A detailed response to the Editor's and Reviewers' comments,

(2) The revised manuscript with tracked changes,

(3) The revised manuscript (clean),

(4) Updated Supplementary Materials.

Should you require any further information or clarification, please do not hesitate to contact us.

With kind regards,

Xiaosheng Qin (Corresponding author)

**Response to Editor**

**Editor Comment:** Both reviewers acknowledged the importance of the topic, but they also raise critical issues regarding the clarifications of the novelty, methodology and results. Please upload the revised manuscript with the point-by-point response.

**Authors' Response:** We thank the reviewer for handling our manuscript and acknowledging the importance of the topic. We have made several revisions that clarify the novelty, methodology and results. Here, we respond to the specific review comments point-by-point (the line, page, figure, and equation numbers correspond to those in the Clean Version).

**Response to Reviewer 1**

**General Comments:**

The authors did a thorough analysis of the future droughts in Monsoon-dominant Asian under the worst-case emission scenario of SSP5-8.5. The analysis of propagation from meteorological to hydrological droughts is new, and the use of bivariate copula function for analyzing drought propagation and spatial concurrence is interesting. I only have a few comments as listed below.

**Authors' Response:** We thank the reviewer for taking time to review our work and for providing valuable comments to improve the manuscript. We have tried to respond to each of the comments and will incorporate the suggested changes in the revised version. Addressing these comments has surely enhanced the quality of the manuscript immensely.

**Specific comments:**

**Reviewer Comment #1:** Abstract: since the propagation from meteorological drought to agricultural drought is a highlight of this work. It's helpful to indicate in the abstract that meteorological and agriculture droughts are measured using SPI and SSI, respectively.

**Authors' Response #1:** We agree that indicating the indices of SPI and SSI explicitly in the abstract is necessary. Accordingly, we have added the following sentence in the abstract [Lines 12 to 13]:

**Lines 12 to 13 (Page 1):** *"Standardized Precipitation Index (SPI) and Standardized Soil moisture Index (SSI) are used to analyse meteorological and agricultural droughts, respectively."*

**Reviewer Comment #2:** Line 61: I believe there are quite a few studies on drought analysis under climate change. I am not sure whether "only" is the most accurate or appropriate term in this case. Citing these works can help readers better understand the current state of research on this topic.

**Authors' Response #2:** We regret this oversight. There are indeed some seminal works on the propagation of meteorological to agricultural droughts. Accordingly, we have revised the sentence to clarify that existing drought propagation studies generally do not examine the repercussions of propagation, specifically the persistence and concurrence of agricultural droughts. Additionally, we have now cited the recent seminal works on meteorological to agricultural drought propagation (Dai et al., 2022; Ding et al., 2021; Fawen et al., 2023; Xu et al., 2023). Accordingly, we have included the following revised sentence for clarification [Lines 70 to 75]:

**Lines 70 to 75 (Page 4):** "*Several seminal works (Dai et al., 2022; Ding et al., 2021; Fawen et al., 2023; Xu et al., 2023) focus on the propagation process, but tend to overlook its repercussions, namely the persistence and spatial concurrence of agricultural droughts. While propagation analysis deals with the transition from a precursor meteorological to an agricultural drought, persistence and spatial concurrence capture their temporal and spatial extents. Given the spatiotemporal nature of droughts, it is crucial to examine these three aspects together for a more comprehensive assessment..*"

**Reviewer Comment #3:** Lines 155-160, a common practice in climate impact studies is to use bias correction techniques and correct the biases in GCM output before any further analysis. Do you think this can help reduce the errors in Figure 2c and Fig. S1? How about the difference between observation and GCM output in Fig. S2?

**Authors' Response #3:** We thank the reviewer for bringing this up. We agree that bias correction can improve the data quality by minimizing errors. However, since this work focuses on drought propagation, it is essential to preserve the interrelationship between precipitation and soil moisture. We refrained from using traditional univariate bias correction techniques, as correcting individual variables (precipitation and soil moisture) could distort the time lags involved in propagation and weaken their correlation.

That said, multivariate bias correction techniques, as explored in recent studies, could be helpful for analysing future drought propagation (Dieng et al., 2022). These methods preserve the inherent relationships between corrected variables, which is crucial for studying extreme events driven by multiple factors (Zscheischler et al., 2019). For example, Meng et al. (2022) applied multivariate bias correction between precipitation and temperature to analyse compound dry and hot events. Similarly, applying such techniques to precipitation and soil moisture could enhance the study of drought propagation dynamics. In the revised manuscript,

we have explicitly acknowledged the limitation of not employing a multivariate bias correction approach in the current study. **Please refer to Lines 540 to 546 on Page 30.**

Nonetheless, the Multi-Model Ensemble (MME) used here still performs well. In response to the reviewer's suggestion, we have included monthly plots showing the mean difference between GCM data (Multi Model Mean using Bayesian Model Averaging) and GLDAS (Observed data). We have also incorporated maps comparing drought properties from historical GCM data and GLDAS. The descriptions of the included thematic maps are as follows [Lines 233 to 239]:

**Lines 233 to 239 (Page 10):** "*Furthermore, monthly plots of the mean differences between GCM data (Multi-Model Mean using Bayesian Model Averaging) and GLDAS (Observed data) show that the median monthly precipitation difference is within ±10 mm for most months across all regions (Fig. S4a). Similarly, the median soil moisture differences remain within ±20 mm across all regions and months, substantially smaller compared to the observed median monthly soil moisture values, which are, around 500 mm (Fig. S4b). Notably, the differences in cumulative drought severities between the observed data and the historical MME indicate that most regions exhibit differences within ±10% for both meteorological (except for certain grids in Western SAS and TIB regions with ±20% difference) and agricultural droughts (Fig. S5).*"

[Figure]

**Figure S4: (a) Monthly distribution of the mean difference between GCM data (Multi-Model Mean using Bayesian Model Averaging) and GLDAS (Observed data) for precipitation and soil moisture, (b) Monthly median of observed precipitation and soil moisture data across four demarcated regions.**

[Figure]

**Figure S5: Differences in cumulative drought severities between the observed and historical MME for (a) meteorological and (b) agricultural droughts in the period between 1975-2014. Cumulative severity is the sum of index values below -0.5 in the period between 1975-2014 for each grid.**

**Reviewer Comment #4:** Line 245: why is R^2 the only performance metrics for soil moisture prediction? With some R^2 values lower than 0.5 in the results, how to justify the accuracy of the RF model or the reliability of its feature importance results?

**Authors' Response #4:** We have added a map showing RMSE values (Fig. S8) to complement the existing $R^2$-themed maps in validating the random forest (RF) models. RMSE (standardized soil moisture) values are much lower (less than 0.1) in the TIB region across timeframes (Fig. S8 in the Supplementary Materials). RMSE values are higher (around 0.7) in SAS and southern EAS in the historical and far-future timeframes, respectively. The above-mentioned statements are mentioned in **Lines 329 to 332** on **Page 16**.

[Figure]

**Figure S8: Performance evaluation (RMSE values) of Random Forest (RF) models for each grid across timeframes.**

These RF models predict time series of soil moisture using temporally varying climatic predictors to understand their influence. A total of 5,364 (1,788 grids x 3 timeframes) RF models were developed, one for each grid across three timeframes. Apart from the climatic

variables, other grid-specific predictors such as elevation (Zhang et al., 2024) and climate characteristics (Zhang et al., 2021), which play a crucial role, are not included in these RF models. Since these variables are temporally constant for a given grid, they do not affect the time series prediction. Consequently, at certain grid points with low $R^2$ values, factors beyond the selected climate variables may influence soil moisture. Despite low $R^2$ values in some cases, the variable with the highest feature importance still demonstrates a relatively stronger influence to the other climate predictors. The above-mentioned statements are mentioned in **Lines 337 to 342** on **Page 16**.

To complement these temporal RF models, which help identify key climatic drivers of soil moisture, spatial RF models have also been developed. These incorporate grid-specific predictors, such as elevation and climate classification, along with aggregated climate variables to predict propagation probability. Please refer to **Author's Response #5** for more details on these spatial RF models.

**Reviewer Comment #5:** Line 250: The predictors that are important in the RF model, are the ones that are important for the estimation of soil moisture. Are they necessarily the same as the ones that may lead to soil moisture deficit? How could the feature importance results be best interpreted?

**Authors' Response #5:** We thank the reviewer for the insightful comment. We understand the concern that the important variables identified in the temporal RF models predicting soil moisture do not explicitly relate to soil moisture deficit. To address this, and to complement the grid-specific temporal RF models, we have included spatial RF models that directly predict drought propagation probabilities (P(SSI<-0.5|SPI<-0.5)) for each timeframe (three RF models in total). These spatial models aim to address the reviewer's concern by using soil moisture deficit, expressed as a conditional probability, as the predictand. The detailed methodology for the spatial RF model is mentioned in Lines 136 to 159, as follows:

**Lines 136 to 159 (Page 6):** *"Hu et al. (2024) developed spatial RF models to assess the relative importance of variables influencing drought propagation. In the present study, these models are considered as spatial because each grid cell is associated with a single propagation probability value (the predictand), forming one row in the training dataset. The predictors include elevation, climate zones, season (represented by month), and the monthly means of various climate variables (i.e. temperature:$T_{Mean}$, precipitation: $Pr_{Mean}$, humidity: $H_{Mean}$, vegetation cover: $VC_{Mean}$, solar radiation at 100 m: $SR_{Mean}$, and wind speed: $W_{Mean}$). The spatial RF model ($f_2$) predicting propagation probability (CP) is expressed as follows:*

[revised manuscript text omitted]

**Technical corrections:**

**Reviewer Comment #6:** Line 100: the "+" sign suggests a summation of these variables inside the function, which is not a rigorous expression. Since SM depends on these variables separately, the notation f(T, Pr, H, VC, SR100, W, TS) would be more appropriate.

**Authors' Response #6:** We thank the reviewer for this comment and agree that it is more appropriate for the RF model to be denoted as SM = f(T, Pr, H, VC, SR, W, TS), instead of using "+" sign. We have incorporated the changes for the RF models in equations 2 and 3. Please refer to **Eq. 2 and 3 on Pages 5 and 6**, respectively.

**General Comments:**

The authors investigate the impacts of climate change on drought propagation from meteorological to agricultural droughts in monsoon-dominant Asian regions. Understanding drought propagation mechanisms under climate change is crucial, particularly in assessing temporal transitions in drought propagation, persistence, and spatial concurrence.

**Authors' Response:** First of all, we thank the reviewer for taking time to review our article. The reviewer has raised several valid and pertinent questions, which we have addressed in our response. We believe that the clarifications and modifications incorporated in the revised version have significantly improved the manuscript.

**Specific comments:**

**Reviewer Comment #1:** The manuscript does not adequately justify the need to study drought propagation, persistence, and spatial concurrence together. Additionally, the claim that studies on meteorological-to-agricultural drought propagation are lacking is inaccurate, as several previous studies already exist (Ding et al., 2021; Dai et al., 2022; Fawen et al., 2023; Xu et al., 2023). Lastly, since this study also focuses on a regional scale, the authors should explicitly clarify its novel contributions and regional significance.

**Authors' Response #1:** We appreciate the reviewer for bringing this up and would like to clarify the overall structure and contribution of the work. The first part of the study focuses on understanding the propagation from meteorological to agricultural droughts using temporal lags and propagation strengths. This analysis helps in forecasting impending agricultural droughts, thereby aiding preparedness efforts. After establishing the relationship between precursor meteorological and antecedent agricultural droughts, the subsequent sections on drought persistence and cross-regional concurrence shift the focus to agricultural droughts. This is because soil moisture deficits (i.e., agricultural droughts) directly impact crop and vegetation growth (Modanesi et al., 2020).

Regarding the novel contributions of this study, while most existing works focus on the propagation process, they often do not examine the subsequent persistence and spatial concurrence of agricultural droughts. In contrast, the present study aims to analyse the persistence and spatial concurrence of agricultural droughts (using SSI directly) and explore their potential interrelationship with the initial propagation process. Our work attempts to access these crucial aspects holistically, rather than examining them in isolation. We apologize for the earlier oversight in stating that studies on meteorological-to-agricultural drought propagation are lacking. We acknowledge that several seminal works do exist on this topic. We have rephrased the relevant sentence to indicate that existing studies primarily focus on the propagation process and tend to overlook its repercussions, namely the persistence and spatial

concurrence of agricultural droughts. We have also cited the relevant seminal works on meteorological-to-agricultural drought propagation as mentioned by the reviewer (Dai et al., 2022; Ding et al., 2021; Fawen et al., 2023; Xu et al., 2023).

Additionally, the novel contributions of the article have now been updated in **Lines 69 to 80 on Page 4**.

**Lines 69 to 80 (Page 4):** *"Despite extensive research on drought-related topics, several gaps remain to be addressed. First, various aspects of droughts, such as propagation, persistence, and spatial concurrence, have largely been studied in isolation. Several seminal works (Dai et al., 2022; Ding et al., 2021; Fawen et al., 2023; Xu et al., 2023) focus on the propagation process, but tend to overlook its repercussions, namely the persistence and spatial concurrence of agricultural droughts. While propagation analysis deals with the transition from a precursor meteorological to an agricultural drought, persistence and spatial concurrence capture their temporal and spatial extents. Given the spatiotemporal nature of droughts, it is crucial to examine these three aspects together for a more comprehensive assessment. Secondly, previous studies have considered drought persistence as a phenomenon spanning two consecutive seasons. However, given the importance of sufficient soil moisture across three sequential seasons (pre-monsoon, monsoon, and post-monsoon) for crop growth, a trivariate extension of the existing bivariate copula framework could more accurately capture drought persistence. Finally, most studies on drought persistence and spatial concurrence reply on observational data. Incorporating climate projection data can offer insights into how droughts may persist and concur spatially in the future under climate change."*

The regional significance of the study has now been mentioned in Lines 83 to 86 on Page 4.

**Lines 83 to 86 (Page 4):** *"The monsoon-dominant Asian region is home to several global rice bowls and wheat baskets that support local food security and play a key role in the international food trade (Gaupp et al., 2020). Given the region's significance, it is appropriate to study a comprehensive drought framework within this study area."*

**Reviewer Comment #2:** Unclear Monsoon-Based Classification for Drought Persistence: The rationale for categorizing drought persistence into pre-monsoon, monsoon, and post-monsoon periods is not well-explained. It remains unclear whether this classification is tied to SPI-SSI propagation dynamics or solely based on SSI thresholds (e.g., SSI < -0.5). A stronger theoretical or empirical basis for this approach is needed. Justify the focus on monsoonal seasons for drought propagation implications.

**Authors' Response #2:**

We deeply appreciate the reviewer's concern on the monsoon-based classification, and offer the following justification:

The rationale for categorizing seasons to analyse drought persistence is mentioned at different sections of the manuscripts such as in Lines 49 to 65 on Page 3 and Lines 168 to 172 on Page 7.

**Lines 49 to 65 (Page 3):** *"After establishing the relationship between precursor meteorological and antecedent agricultural droughts, the persistence and cross-regional concurrence of agricultural droughts need to be analysed. Ford and Labosier (2014) defined persistence as the tendency of droughts to extend temporally from one season to another. Inter-seasonal drought dynamics can also be analysed using a copula-based multivariate approach, treating drought index values from each season as random variables (Chen et al., 2016; Fang et al., 2019; Shi et al., 2020; Swain et al., 2024; Xiao et al., 2017). These previous studies analysed transition from dryness to wetness (and vice versa), prolonged wetness and prolonged dryness between two successive seasons using bivariate copulas. Of the four possible inter-seasonal scenarios, prolonged dryness, referring to inter-seasonal drought persistence, can affect vegetation and crop growth. Nonetheless, limiting the analysis to two consecutive seasons may overlook long-term drought persistence across seasons that overlap with crop growth cycles. Crops are cultivated across multiple seasons throughout the study area, with different stages of crop growth aligning with the pre-monsoon, monsoon, and post-monsoon seasons. For example, in China, spring crops are typically sown in May and harvested around October, while winter crops like wheat are planted in September and harvested by following June (Li and Lei, 2021). Similarly, in South Asia, different crops are grown in three different phases including Zaid, Kharif, and Rabi, that correspond to the pre-monsoon, monsoon, and post-monsoon seasons, respectively (Joseph and Ghosh, 2023). Adequate soil moisture during all these periods is critical for crop development, and agricultural droughts affecting these three phases could be highly detrimental."*

**Lines 168 to 172 (Page 7):** *"While monsoonal droughts are a key focus, given that these months account for the majority of annual rainfall, droughts during the pre- and post-monsoon periods can also severely impact crop growth. Yang et al. (2021) highlight that soil moisture deficits prior to planting can impair seedling root development, significantly affecting crop yields. Thus, droughts in pre-monsoon that coincide with land preparation and early sowing stages must also be analysed. Moreover, residual soil moisture from the monsoon is crucial for winter (or post-monsoon) crop growth."*

Drought persistence in this study is calculated directly using a soil moisture-based index (SSI at a monthly timescale), which reflects agricultural droughts. It is not derived from SPI values (meteorological droughts) or drought propagation probability (defined by Conditional Probability, CP). However, as shown in Fig. 12, there is a strong linear relationship between drought persistence (defined by Joint Probability, JP) and drought propagation (CP). In Fig. 12, changes in CP values between future and historical timeframes were compared with the corresponding JP values across all grids using scatterplots and thematic maps. This highlights regions with high drought persistence. The strong correlation (approximately 0.85, $p < 0.001$) suggests that grids experiencing accelerated drought propagation are also likely to face

persistent droughts across seasons in the future, and vice versa. Please refer to **Lines 495 to 501 on Page 28** for the above-mentioned statements.

[Figure]

**Figure 12: (a) Relationship between drought propagation and persistence using changes in their probability values between future and historical timeframes, and (b) spatial maps showing bivariate classification based on future persistence and propagation values to identify vulnerable grids.**

Further details explaining the rationale for explicitly studying drought persistence and spatial concurrence using SSI values are provided in our next response (**Authors' Response #3**). We kindly refer the reviewer to that section.

**Reviewer Comment #3:** Ambiguity in Drought Concurrence Analysis: The assessment of spatial drought concurrence relies on SSI thresholds but does not explicitly link to SPI-driven propagation. The authors should clarify whether the observed concurrence reflects independent agricultural droughts or is influenced by meteorological drought propagation.

**Authors' Response #3:** We thank the reviewer for the observation and will clarify the ambiguity regarding the drought concurrence analysis. As mentioned in earlier responses, the persistence and spatial concurrence of agricultural droughts are not explicitly based on SPI-driven propagation measures in their methodological computation. However, the resulting

patterns from both persistence and spatial concurrence analyses are compared with propagation probabilities to understand their influence. The following revisions are provided in the revised manuscript for clarification:

**Lines 477 to 484 (Page 26):** *"Although SPI-driven propagation values were not directly used to calculate the spatial concurrence of agricultural droughts (only SSI value were considered), several key findings align with those from the propagation analysis. For instance, concurrent cross-regional droughts between South Asia (SAS) and East Asia (EAS) are projected to increase in the far-future timeframe (Fig. 11). In contrast, Southeast Asia (SEA) is expected to experience more non-synchronous (i.e., decreased concurrence) droughts with SAS and EAS in the future compared to the historical period. These findings are consistent with the trends in propagation probabilities: both SAS and EAS show increased propagation from meteorological to agricultural droughts in the future (Fig. 4(a)), while SEA shows a decline in propagation probability in the far-future relative to the historical timeframe. Thus, the results from the propagation and concurrence analyses are in agreement."*

[Figure]

**Figure 11: Spatial concurrent return periods between region pairs across timeframes, with random variables (percentage of area under drought annually) shown as black dots.**

[Figure]

**Figure 4: (a) Comparison of propagation probabilities across four regions, seasons, and timeframes.**

**Rationale for using agricultural droughts (SSI) directly to analyse drought persistence and spatial concurrence:**

The reviewer's 2nd and 3rd comments raise a common concern about whether the agricultural drought indices used in analysing drought persistence and concurrence are directly influenced by meteorological drought propagation. To address this, we computed the probability of meteorological droughts occurring under the condition of existing agricultural droughts (P(SPI<0|SSI<-0.5)) across three timeframes, and compared it with the forward propagation probability (P(SSI<0|SPI<-0.5)) shown in **Fig. 7**. This reverse propagation probability, (P(SPI<0|SSI<-0.5)), reflects the extent to which an agricultural drought is linked to a prior meteorological drought. The density plots show a decline in reverse propagation probability values in both the near- and far-future scenarios relative to the historical period across all regions, with the most significant decrease observed in EAS in the far-future timeframe. This suggests that the number of agricultural drought events not directly attributable to meteorological droughts is expected to rise.

Simultaneously, the forward propagation probability (P(SSI<0|SPI<-0.5)) indicates that meteorological droughts are increasingly driving agricultural droughts in SAS and EAS in the future (Fig. 4). Thus, while meteorological-driven agricultural droughts are projected to increase, so too are those unrelated to rainfall deficits. Random forest models used to predict soil moisture further reveal that temperature becomes the dominant driver of agricultural droughts across more than 50% of the study area in future scenarios, compared to about 25% in the historical period (Fig. 5(c) in the revised version). This supports the observed increase

in non-rainfall-related agricultural droughts. Under the SSP5-8.5 scenario, significant temperature increases are expected towards the end of the century (Qiao et al. 2023). This warming could intensity soil moisture deficits, leading to a shift from meteorological-driven to temperature-driven agricultural droughts. Additionally, above-average rainfall (SPI>0) may occur in the form of short-term, intense storms, which may not adequately replenish soil moisture under higher temperatures, which is a situation exacerbated by climate change. These findings support the methodological decision to use SSI directly to assess drought persistence and concurrence, rather than relying solely on SPI. In EAS, and SAS, although meteorological-driven agricultural droughts are projected to rise in the far-future, they are likely to be more severe due to the increasing influence of temperature, potentially resulting in more frequent compound drought-heatwave events.

We hope these new insights address the methodological concerns raised and justify the direct use of SSI in our analysis. We have incorporated these findings in the revised manuscript in **Section 4.1.4, Lines 385 to 403, Page 20.**

[Figure]

**Figure 7: Reverse propagation probability (P(SPI<0|SSI<-.5)) indicates if an agricultural drought is linked to a prior meteorological drought.**

**Reviewer Comment #4:** L93: Random Forest Model Design: The use of soil moisture as the predictand (rather than drought propagation metrics) limits the model's ability to identify key drivers of propagation. Restructuring the model to treat propagation as the predictand (with climatic variables as predictors) would better address the study's primary objective.

**Authors' Response #4:** We thank the reviewer for the insightful comment. We understand the concern that the important variables identified in the temporal RF models predicting soil moisture do not explicitly relate to drought propagation. To address this, we have incorporated spatial RF models aimed at directly predicting drought propagation probabilities (P(SSI<-0.5|SPI<-0.5)) for each timeframe (i.e., three separate RF models). These spatial RF models are designed to predict propagation probabilities at each grid, with each grid contributing one training data row per model. We have retained the temporal RF models as they offer complementary insights. More methodological details of the updated spatial RF models are described in Lines 136 to 159 on Page 6.

**Lines 136 to 159 (Page 6):** *"Hu et al. (2024) developed spatial RF models to assess the relative importance of variables influencing drought propagation. In the present study, these models are considered as spatial because each grid cell is associated with a single propagation probability value (the predictand), forming one row in the training dataset. The predictors include elevation, climate zones, season (represented by month), and the monthly means of various climate variables (i.e. temperature:$T_{Mean}$, precipitation: $Pr_{Mean}$, humidity: $H_{Mean}$, vegetation cover: $VC_{Mean}$, solar radiation at 100 m: $SR_{Mean}$, and wind speed: $W_{Mean}$). The spatial RF model ($f_2$) predicting propagation probability (CP) is expressed as follows:*

[revised manuscript text omitted]

**Minor comments:**

**Reviewer Comment #5:** L13-L16: " In terms of the return period, all-season droughts that historically occurred once in more than 50 years could happen as frequently as every five years by the far-future (2061-2100) at the hydrologically significant Tibetan Plateau." The statement is confusing. Rephrase for clarity.

**Authors' Response #5:** We thank the reviewer for the comment and would like to rephrase the statement as follows:

**Lines 16 to 18 (Page 1):** *"At the hydrologically significant Tibetan Plateau, all-season droughts that were historically rare, with return periods exceeding 50 years, could occur as frequently as once every 5 years in the far-future period (2061-2100)."*

**Reviewer Comment #6:** L17-L19: "The spatial concurrence of monsoonal agricultural droughts between region pairs such as South Asia (SAS), East Asia (EAS), Southeast Asia (SEA), and Tibetan region (TIB) was also assessed." Replace methodological descriptions (e.g., region pairs assessed) with concrete findings.

**Authors' Response #6:** We thank the reviewer for this comment and will replace this statement based on methodology with more appropriate findings in the abstract in **Lines 19 and 20 on Page 1.**

*"The increasing non-rainfall-related agricultural droughts in the far-future could be attributed to the rise in temperature."*

**Reviewer Comment #7:** L27: Cite specific literature comparing disaster impacts.

**Authors' Response #7:** We have added appropriate literatures for the specified statement in the revised version.

**Lines 27 and 28 on Page 2: "***Droughts, though not instantaneous, rank among the costliest disasters due to their prolonged and widespread impacts (Hao et al., 2016; Smith and Katz, 2013; Smith and Matthews, 2015)."*

The cited works (Smith and Katz, 2013; Smith and Matthews, 2015) compare different disasters and their economic impacts.

**Reviewer Comment #8:** L40-41: "While comparing propagation between basins of different climate zones, Zhang et al. (2021) found arid basin to have lower propagation durations compared to humid and sub-humid basins" Rephrase for clarity.

**Authors' Response #8:** We thank the reviewer for the suggestion and have rephrased the sentence for improved clarity as follows:

**Lines 41 to 43 on Page 3:** *"Climate characteristics are known to influence drought propagation. For instance, Zhang et al. (2021) found that arid basins tend to have shorter propagation durations than humid and sub-humid basins."*

**Reviewer Comment #9:** L64-65: "To address the aforementioned gaps, this study proposes a comprehensive copula-based multivariate probabilistic approach, utilizing climate model projection data." Separate the copula method and climate model applications to avoid logical gaps.

**Authors' Response #9:** We thank the reviewer for the suggestion and have rephrased the sentence for improved clarity as follows:

**Lines 81 to 82 (Page 4)***: "To address the aforementioned gaps, this study proposes a comprehensive copula-based multivariate probabilistic framework, which will be applied to climate model projection data."*

**Reviewer Comment #10:** 2.1.1: Describe monthly precipitation, SPI/SSI calculations, and soil moisture datasets.

**Authors' Response #10:** We thank the reviewer for the comment, and we have included a description as follows:

**Lines 97 to 107 (Pages 4 and 5)***: "SPI is based on monthly precipitation, while SSI relies on monthly soil moisture. These standardized indices are widely applied in drought studies due to their flexibility in capturing seasonal variations through different timescales (TS). Timescales represent accumulation periods, defined by aggregating precipitation or soil moisture over a specified number of months. For each month, the accumulated values (for a given timescale) are transformed into a standardized index. The standardization process involves the computing the cumulative probability of the accumulated values for a given month, which is then transformed to a standardized normal variate (Z-score) with a mean of zero and a standard deviation of one. This process is repeated for all the months (McKee et al., 1993). Standardization enables fair comparison of drought conditions across the diverse study area. Standardized values below (above) zero indicate dry (wet) conditions. The Pearson correlation between monthly SSI (at a one-month timescale) and SPI across various timescales (TS = 1 to 12) is calculated for each month. The SPI timescale that shows the highest correlation with SSI indicates the drought propagation duration for that month (Barker et al., 2016). Details of the precipitation and soil moisture data used to compute these indices are provided in Section 3."*

**Reviewer Comment #11:** 2.1.3: Please provide a detailed description of the hyperparameters in the random forest model (e.g., the number of trees, maximum depth), explaining the parameter tuning process to validate the robustness of the model. Furthermore, it is recommended to include other evaluation metrics (e.g., RMSE) to more accurately demonstrate model performance. Please provide the cross-validation of RF models and shows the R2 of different regions.

**Authors' Response #11:** We appreciate the reviewer's suggestion regarding hyperparameter tuning and performance evaluation of the RF models. Due to the computational intensity involved in training 5364 (1788 grids X 3 timescales), we adopted a fixed set of fixed hyperparameters similar to the ones used by Dai et al. (2022). The hyperparameters are listed as follows:

- ntree = 500 (number of trees)
- mtry = 5 (number of variables tried at each split)
- nodesize = 5 (minimum size of terminal nodes)

We have added a map showing RMSE values (Fig. S8) to complement the existing $R^2$-themed maps (Fig. 5a) in validating the random forest (RF) models. RMSE (standardized soil moisture) values are much lower (less than 0.1) in the TIB region across timeframes (Fig. S8 in the Supplementary Materials). RMSE values are higher (around 0.7) in SAS and southern EAS in the historical and far-future timeframes, respectively. The above-mentioned statements are mentioned in **Lines 327 to 330** on **page 16**.

[Figure]

**Figure S8: Performance evaluation (RMSE values) of Random Forest (RF) models for each grid across timeframes.**

An extensive grid-based hyperparameter tuning and evaluation during the testing phase are done for the spatial RF models. Please refer to our response to **Reviewer Comment #4.**

**Reviewer Comment #12:** Fig.2b: Provide citations for the basis of regional divisions, which is crucial to spatial concurrence analysis.

**Authors' Response #12:** The regional divisions are based on thematic maps from works such as Giorgi and Bi (2009) and Sillmann et al. (2013). The citations have been added appropriately in the revised version.

**Reviewer Comment #13:** Fig. 2c: Compare GLDAS and GCM-derived SPI monthly drought characters (not average precipitation) to validate GCM reliability for drought propagation.

**Authors' Response #13:** We thank the reviewer for the insightful comment. We have prepared thematic maps of cumulative drought severities from GLDAS and GCM data.

[Figure]

**Figure S5: Differences in cumulative drought severities between the observed and historical MME for (a) meteorological and (b) agricultural droughts in the period between 1975-2014. Cumulative severity is the sum of index values below -0.5 in the period between 1975-2014 for each grid.**

**Lines on 237 to 239 Page 10**: *"Notably, the differences in cumulative drought severities between the observed data and the historical MME indicate that most regions exhibit differences within ±10% for both meteorological (except for certain grids in Western SAS and TIB regions with ±20% difference) and agricultural droughts (Fig. S5)."*

**Reviewer Comment #14:** Datasets: Replace GLDAS precipitation/soil moisture with more reliable datasets (e.g., MSWEP for precipitation, GLEAM for soil moisture).

**Authors' Response #14:** MSWEP and GLEAM (with MSWEP as the major input) have been commonly used in drought propagation studies to minimize uncertainty arising from disparate data sources (Gupta and Karthikeyan, 2024; Odongo et al., 2023). For a similar reason, ensuring consistency between precipitation and soil moisture data, GLDAS was used in this study. GLDAS datasets are widely employed in drought research and have been shown to effectively capture major historical drought events. For instance, we plotted the percentage of areas affected by meteorological and agricultural droughts using GLDAS data (**Fig. S6**), which clearly reflects significant events such as the 2009 drought (Barriopedro et al., 2012).

Additionally, Gupta and Karthikeyan (2024) reported good agreement in meteorological drought characteristics across MSWEP, CHIRPS, and GLDAS, supporting the reliability of GLDAS. While MSWEP and GLEAM are excellent datasets for drought propagation studies, existing literature suggests that the GLDAS performs comparably well. Furthermore, the three timeframes used in the study, historical (1975-2014), near-future (2021-2060, and far-future (2061-2100), were selected to ensure equal-length epochs of 40 years for consistent comparison of drought propagation, persistence, and spatial concurrence. However, MSWEP and GLEAM begin only in 1979 and 1980, respectively, making them unsuitable for this time range. Given

these considerations, we believe GLDAS is an appropriate choice for the present study. The above explanations have been mentioned in **Lines 244 to 254 on Pages 10 and 11.**

[Figure]

**Figure S6: The percentage of drought-affected areas (based on the percentage of grids falling at different ranges of SPI and SSI values) annually. Spatial maps of 2009 drought events.**

**Cited References**

[revised manuscript text omitted]